# An epithelial signalling centre in sharks supports homology of tooth morphogenesis in vertebrates

Alexandre P Thiery[1,2], Ariane SI Standing[3], Rory L Cooper[1,4], Gareth J Fraser[3]*

[1]Department of Animal and Plant Sciences, University of Sheffield, Sheffield, United Kingdom; [2]Department of Craniofacial Development and Stem Cell Biology, King's College London, London, United Kingdom; [3]Department of Biology, University of Florida, Gainesville, United States; [4]Department of Genetics and Evolution, University of Geneva, Geneva, Switzerland

**Abstract** Development of tooth shape is regulated by the enamel knot signalling centre, at least in mammals. Fgf signalling regulates differential proliferation between the enamel knot and adjacent dental epithelia during tooth development, leading to formation of the dental cusp. The presence of an enamel knot in non-mammalian vertebrates is debated given differences in signalling. Here, we show the conservation and restriction of *fgf3*, *fgf10*, and *shh* to the sites of future dental cusps in the shark (*Scyliorhinus canicula*), whilst also highlighting striking differences between the shark and mouse. We reveal shifts in tooth size, shape, and cusp number following small molecule perturbations of canonical Wnt signalling. Resulting tooth phenotypes mirror observed effects in mammals, where canonical Wnt has been implicated as an upstream regulator of enamel knot signalling. In silico modelling of shark dental morphogenesis demonstrates how subtle changes in activatory and inhibitory signals can alter tooth shape, resembling developmental phenotypes and cusp shapes observed following experimental Wnt perturbation. Our results support the functional conservation of an enamel knot-like signalling centre throughout vertebrates and suggest that varied tooth types from sharks to mammals follow a similar developmental bauplan. Lineage-specific differences in signalling are not sufficient in refuting homology of this signalling centre, which is likely older than teeth themselves.

*For correspondence:
g.fraser@ufl.edu

Competing interest: The authors declare that no competing interests exist.

## Editor's evaluation

The manuscript by Thiery et al brings valuable results for understanding the developmental strategies of signalling centres during shark teeth appearance. We believe that it will bring new insight into the general understanding of the phylogenesis of organizers or signalling centres.

## Introduction

Diversification of the dentition has been instrumental in the success of vertebrates. The dentition has become highly adapted to its respective environmental niche, and as a result has given rise to a plethora of unusual dental forms (*Evans et al., 2007*; *Fraser et al., 2008*; *Hulsey et al., 2020*; *Jernvall and Thesleff, 2012*; *Kolmann et al., 2019*; *Thiery et al., 2017*; *Tucker and Fraser, 2014*). It is generally observed that there is an overall increase in dental morphological complexity throughout evolution, culminating in mammals which possess multiple regionalised tooth types (heterodonty) (*Jernvall and Thesleff, 2012*; *Kavanagh et al., 2007*; *Tucker and Fraser, 2014*). A trade-off between tooth morphological complexity and dental regenerative ability is thought to exist within the mammalian

lineage, with precise occlusion favoured at the expense of dental regeneration (*Jernvall and Thesleff, 2012*). The study of dental development has mostly focussed on transcriptional regulators, which mediate growth, morphogenesis, and cellular differentiation. Dental development is tightly regulated via interactions between the oral epithelium and underlying neural-crest derived mesenchyme (*Zhang et al., 2005*). Canonical Wnt, fibroblast growth factor (Fgf), hedgehog (Hh), bone morphogenetic protein (Bmp), and Notch signalling pathways have all been implicated as important regulators of tooth development and morphogenesis (*Thesleff and Sharpe, 1997*). Importantly, a unifying characteristic for the progression of vertebrate tooth development is the high level of cellular expression of developmental keystone genes (*Hallikas et al., 2021*). This suggests an essential core gene set (*Fraser et al., 2010*; *Rasch et al., 2016*) and their interactive signalling might be robust evolutionarily and highly conserved, capable of shape modifications of a conserved core vertebrate unit – the tooth.

Teeth begin their development from an initial epithelial placode, which proceeds to proliferate and grow in size during the subsequent bud stage. Following the bud stage, the tooth enters the cap stage, which marks the onset of morphogenesis. Morphogenesis of epithelial appendages coincides with a change in shape of the epithelial placode. This process can be driven via differential proliferation rates (i.e., teeth *Jernvall et al., 1994*; *Salazar-Ciudad and Jernvall, 2010*), cell shape changes (i.e., intestinal crypt *Sumigray et al., 2018*), or cell migration (i.e., hair *Ahtiainen et al., 2014*). Signalling molecules drive reciprocal epithelial to mesenchymal signalling and regulate the morphogenesis of epithelial appendages (*Mustonen et al., 2002*; *Thesleff and Sharpe, 1997*).

During morphogenesis, the tooth acquires its defining morphological feature: the dental cusp. In mammals, dental cusps are regulated by an epithelial signalling centre found at the tip of the first cusp, known as the primary enamel knot (EK). The EK is molecularly identifiable as non-proliferative and expresses Wnt, Bmp, Fgf, and Hh markers (*Jernvall and Thesleff, 2012*). Fgf signalling drives differential proliferation between the EK and rapidly proliferating adjacent dental epithelium, leading to folding of the epithelium at the site of dental cusps and the formation of the final tooth shape (*Jernvall et al., 1998*; *Jernvall et al., 1994*; *Salazar-Ciudad and Jernvall, 2010*). During the final stages of dental morphogenesis, the EK dramatically reduces in size as cells undergo apoptosis (*Jernvall et al., 1998*). In multi-cuspid teeth, such as mammalian molars, subsequent signalling centres termed secondary enamel knots (SEK) form at the site of each extra cusp and are thought to be induced by the EK. Folding of the dental epithelium between primary and secondary EKs leads to formation of cervical loops in-between each cusp (*Jernvall et al., 1994*; *Salazar-Ciudad, 2012*). Most of our understanding of tooth shape comes directly from the study of mammalian molar cusp development. Whilst it is known that EK signalling centres are found throughout mammals, their presence in other vertebrates is less clear.

The Wnt/β-catenin signalling pathway appears to govern a variety of events during tooth development from initiation to morphogenesis. Canonical Wnt signalling is active in major cellular contributions of the developing tooth, with varying roles in the mesenchyme and epithelium. For example, epithelial activation of Wnt/β-catenin signalling results in continuous initiation of new tooth units in mice; however, activation of Wnt signalling in the dental mesenchyme has an opposite effect, inhibiting sequential tooth emergence (*Järvinen et al., 2018*). Whilst over a dozen markers have been described within the mammalian EK, Wnt signalling appears to be a primary upstream regulator of EK patterning (*Ahtiainen et al., 2016*; *Järvinen et al., 2018*). Given this association with Wnt signalling and the control of tooth shape, including EK patterning, it seems appropriate to investigate further upstream effects of Wnt perturbation in alternative vertebrate models (*Fraser et al., 2013*; *Richman and Handrigan, 2011*). Lef1-/- mutants exhibit arrested tooth development during bud stage, with the tooth cap and associated cusps failing to both form and express EK markers, including Fgf4, Shh, and Bmp4 (*Kratochwil et al., 2002*). Fgf4 is capable of rescuing tooth development in Lef1-/- mutants (*Kratochwil et al., 2002*). Furthermore, the inhibition of Wnt signalling during early bud stage via ectopic expression of the Wnt antagonist Dickkopf-related protein 1 (Dkk1) leads to blunted cusps and a reduction in Bmp4 signalling (*Liu et al., 2008*). These results suggest that Wnt signalling falls upstream of Fgf and Bmp signalling in the EK. However, more complex feedback loops are involved, as mesenchymal-specific knock-down of Bmp4 (Bmp4ncko/ncko) also leads to a reduction in *Lef1* expression (*Jia et al., 2016*).

In the subset of reptiles studied to date, there appears to be no clear histologically definable EK (*Buchtová et al., 2008*; *Weeks et al., 2013*), although in some species there is a thickening of the

inner dental epithelium which leads to the asymmetric deposition of enamel and the formation of cusps (*Zahradnicek et al., 2014*). Furthermore, whilst in reptiles, cells of the inner dental epithelium are non-proliferative, this region is not as highly restricted to the cusp as in mammalian molars (*Buchtová et al., 2008*; *Handrigan and Richman, 2011*). However, there is conservation of signalling within an EK-like signalling centre (*Richman and Handrigan, 2011*). Similar EK-like signalling centres have been described in teleosts, with chemical perturbation of these signalling pathways resulting in shifts in cusp number (*Fraser et al., 2013*). Sharks, which are basal crown gnathostomes, also possess a variety of tooth types, with clearly defined cusps. In sharks and rays, there is a region of non-proliferative cells within the apical dental epithelium thought to be associated with the primary cusp (i.e., the primary enameloid knot; *Rasch et al., 2020*; *Rasch et al., 2016*), however the presence of a definitive EK has not yet reached a consensus (*Debiais-Thibaud et al., 2015*), and some have even argued that the EK is uniquely a mammalian innovation (*Handrigan and Richman, 2011*; *Handrigan and Richman, 2010*; *Richman and Handrigan, 2011*; *Weeks et al., 2013*).

Sharks and rays (elasmobranchs) represent two major subdivisions of the cartilaginous fishes, and house oral teeth in both the upper and lower jaws, and also possess tooth-like structures embedded within the skin, made of dentine and enamel-like (enameloid) tissues (*Figure 1A*). The diversity in tooth shape in the elasmobranchs is vast, ranging from flattened tooth units associated with crushing prey items, to multi-cuspid and serrated tooth units necessary for cutting (*Jambura et al., 2020*). The presence of multiple cusps as a standard tooth form for most sharks suggests conserved developmental control of tooth cusp morphogenesis across vertebrates. Given the conservation of general tooth development among vertebrates (*Fraser et al., 2004*; *Martin et al., 2016*; *Rasch et al., 2020*; *Rasch et al., 2016*; *Thiery et al., 2017*; *Weeks et al., 2013*), it seems appropriate that the epithelial and mesenchymal signals regulating dental morphogenesis would also be conserved, including the EK signalling centre. This would imply that this crucial tooth signalling centre is older than the mammalian clade. Therefore, due to the vast array of dental phenotypes in sharks and their position in the wider context of vertebrate phylogeny, we employ the shark as a developmental model for non-mammalian tooth morphogenesis. The small spotted catshark (*Scyliorhinus canicula*) has recently become a popular model for the study of developmental biology, including tooth and skin appendage development (*Berio et al., 2021*; *Cooper et al., 2018*; *Cooper et al., 2017*; *Debiais-Thibaud et al., 2015*; *Martin et al., 2016*; *Rasch et al., 2016*). In addition to the oral dentition (*Figure 1B, C*), sharks house a unique set of tooth-like denticles embedded in the skin (*Figure 1A*; *Cooper et al., 2018*), which also display a vast diversity of cusp shapes (*Ankhelyi et al., 2018*; *Gabler-Smith et al., 2021*; *Sibert and Rubin, 2021*).

In order to determine the extent of signalling conservation between mammals and sharks during dental morphogenesis, we have documented the expression of mammalian EK markers in the small-spotted catshark, *S. canicula*. Furthermore, given the importance of Wnt signalling during dental morphogenesis and cusp formation in mammals, we sought to perturb Wnt signalling and determine its function during catshark tooth development. Following small molecule Wnt signalling perturbation, we identify shifts in tooth shape, size, and cusp number, and model the resulting phenotypes in silico using the 'ToothMaker' programme (*Salazar-Ciudad and Jernvall, 2010*; *Savriama et al., 2018*). Our results reveal that despite differences in molecular signalling between the catshark and mammals, the fundamental components of the EK signalling centre are highly conserved. Therefore, EK signalling at the apex of the developing tooth cusp is a vertebrate innovation and contributes to the vast range of vertebrate dental phenotypes.

## Results

### Histological and morphological analysis of the catshark dentition

The first dental generation develops relatively superficially on the oral surface before the dental lamina has fully invaginated (*Figure 1D*). As the dental lamina grows, more generations emerge. New teeth are initiated at the tip of the dental lamina and move anteriorly in a conveyor belt-like manner. As teeth move along this trajectory, they undergo morphogenesis and matrix secretion, before erupting on the oral surface on the labial side of the dental lamina (*Figure 1E*). Adjacent tooth families are staggered in the timing of their initiation (*Figure 1C, J and L*), resulting in differences in the developmental stages of adjacent teeth.

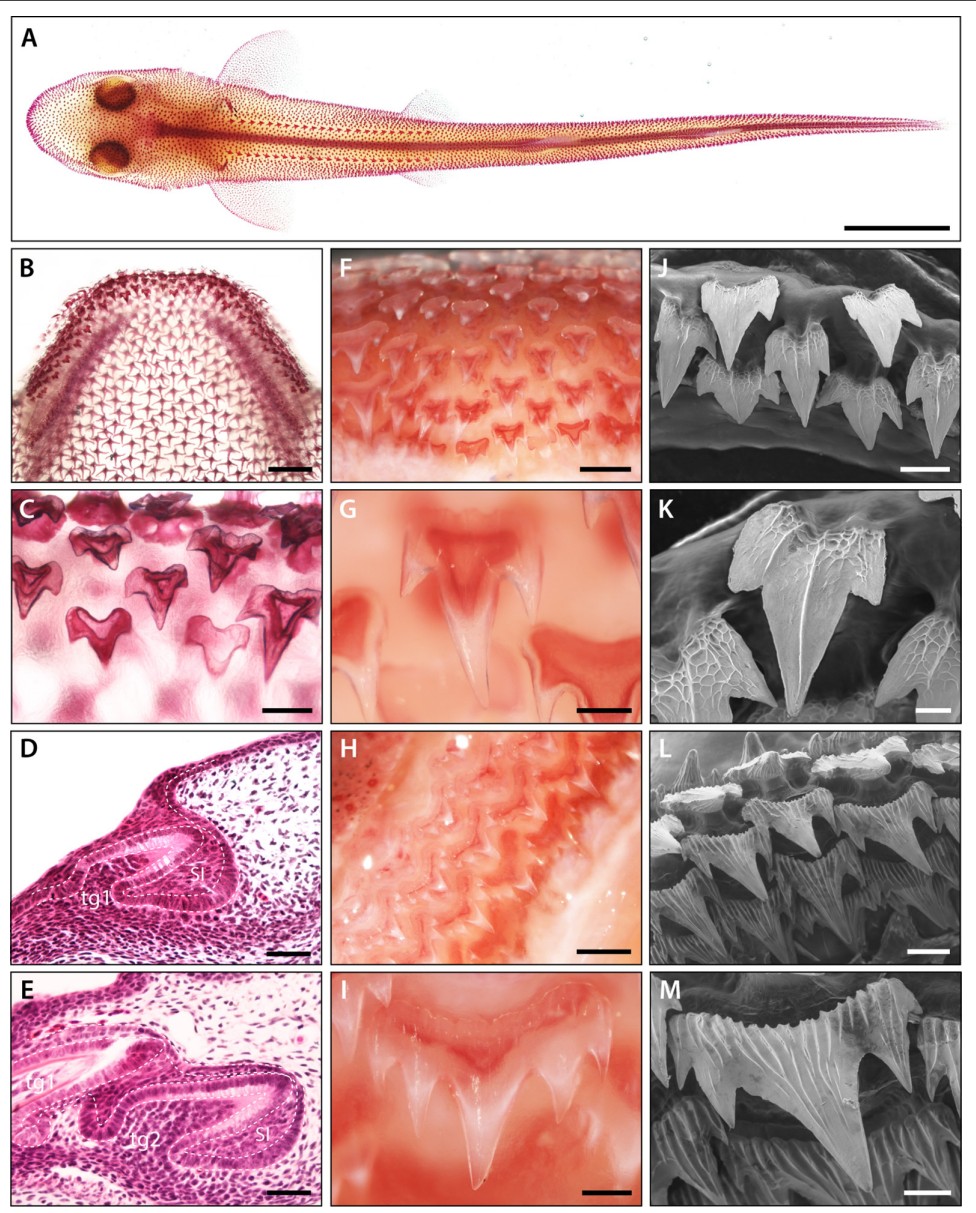

**Figure 1.** Pattern and morphology of the catshark dentition. Images of catshark samples cleared and stained with alizarin red, reveal pattern and morphology of the dentition in the lower jaw (**A–C and F–I**). (**A**) Dorsal view of hatchling stage catshark (stage 34) revealing denticle on the body surface. (**B**) Dorsal view of hatchling stage (stage 34) lower jaw. (**C**) Magnification of B, showing staggered pattern of adjacent tooth families. Histological staining with haematoxylin and eosin on sagittal cross sections through early stage 32 (**D**) and late stage 32 (**E**) lower jaws shows the growth of the DL during dental development. The first dental generation (tg1) develops relatively superficially at the oral surface, before full invagination and elongation of the DL (**D**). The DL then grows deep into the underlying mesenchyme, with the second (tg2) and subsequent dental generations initiated at the SL (**E**). The addition of numerous successional dental generations can be seen in the adult jaw (**F and H**). At the jaw symphysis (**F**), teeth often remain tricuspid (**G**). However, in lateral regions (**H**), teeth develop 5–7 cusps (**I**). Scanning electron microscope (SEM) images reveal tricuspid teeth in the embryo (stage 33) (**J–K**) and pentacuspid teeth in the adult (**L and M**). Scale bars are 10 mm in A; 1 mm in B, F, and H; 250 µm in C, G, I, and M; 50 µm in D, E, and K; 125 µm in J, and 500 µm in L. ora; oral, abo; aboral, lin; lingual, lab; labial.

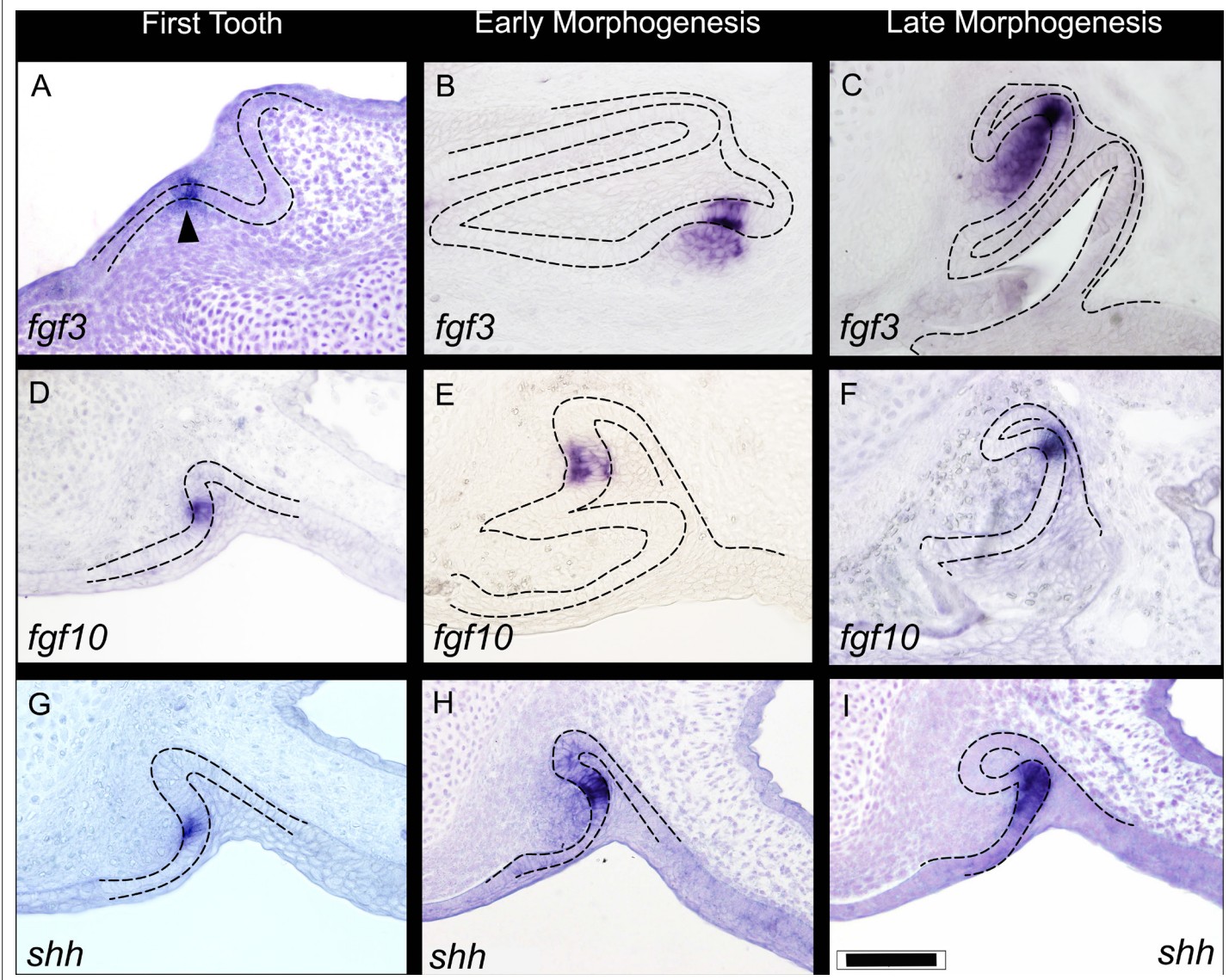

**Figure 2.** Expression of *fgf3*, *fgf10*, and *shh* during three key stages of shark tooth morphogenesis. First tooth initiation (**A,D,G**); early morphogenesis (**B, E, H**); and late morphogenesis (**C,F,I**). First tooth initiation is superficial with expression of the *fgf3*, *fgf10*, and *shh* in a restricted compartment of the dental epithelium; *fgf3* expression extends into the underlying papilla mesenchyme (arrowhead). During early morphogenesis, both *fgf3* and *fgf10* continue to show restricted expression to the site of the EK-like unit (although the underlying mesenchymal expression of *fgf3* does expand further throughout the mesenchymal papilla at later stages). *shh* expression at this stage, however, starts to expand away from the EK-like unit of the dental epithelium (**H**). At late morphogenesis (**C,F,I**), *fgf3* has an even more expanded expression pattern in the mesenchymal papilla, yet the epithelial expression is still very restricted the apex of the tooth cusp, C. *fgf10* expression is always confined to the apical tip (**F**), and *shh* expression expands further throughout the inner dental epithelium in the labial cusp sites (**I**). Scale bar in I=100 µm.

The online version of this article includes the following figure supplement(s) for figure 2:

**Figure supplement 1.** Expression of *bmp4* and *sostdc1* in the shark dentition.

There is an observable shift in cusp number between embryonic and juvenile catshark stages. Embryonic teeth generally develop with 3 cusps (*Figure 1C, J and K*). In contrast, adult catsharks can possess anywhere between 3 and 7 cusps depending on the position of the tooth along the jaw margin (*Figure 1F–I*). Given these changes in cusp number, we aimed to investigate how the development of cusps is regulated. To see if there is a conserved EK signalling centre in the catshark, we documented the expression of known mammalian EK markers during odontogenesis (*Figure 2* and *Figure 2—figure supplement 1*). In situ hybridisation was undertaken on sagittal paraffin sections to observe the expression of EK markers. We used embryonic catshark samples at late stage 32

(~125 days post fertilisation [dpf]) (*Ballard et al., 1993*), as two to three dental generations had already undergone the process of dental initiation by this stage, allowing us to compare early and late stage dental morphogenesis.

## Key markers of tooth morphogenesis in sharks

To identify genes associated with tooth morphogenesis in the shark and to appreciate the level of conservation in gene expression, several markers of tooth initiation and morphogenesis were investigated in the shark. First, we sought to characterise the expression of genes from known pathways (Fgf and Hh) associated with initial patterning of tooth cusps in mice (*Ahtiainen et al., 2016*; *Du et al., 2017*; *Kettunen et al., 2000*). Our results show the expression of three genes, *fgf3*, *fgf10*, and *shh*, all associated with early restricted epithelial expression in the first, superficial generation of teeth (*Figure 2A, D and G*), and later in morphogenesis (*Figure 2B, C, E, F, H1*; *Figure 3G and H*) as tooth shape progresses, with the exception of *fgf3* which is also expressed in the underlying mesenchymal apex of the dental papilla (*Figure 2A–C*). Importantly, the superficial nature of expression in these three genes is associated with the initiation of the first tooth placode (*Figure 2A, D and G*), and this could in fact mark the onset of the primary EK-like signalling in the shark tooth (see Discussion for further comparative information).

Interestingly, the expression of *fgf3* and *fgf10* is almost identical between the shark and the opossum (*Monodelphis domestica*), which shows a subtle shift in expression of *Fgf10* compared to placental mammals, with *Fgf10* expressed in both the EK and the underlying dental mesenchyme in the opossum, and not in the EK of the mouse (*Kettunen et al., 2000*; *Moustakas et al., 2011*). We find *fgf3* expressed in the tip of the dental epithelium, whilst its mesenchymal expression is confined just below the epithelium (*Figure 3G*). The expression of *fgf10* is highly restricted to only a few epithelial cells at the tip of the developing tooth, during cap stage (*Figure 3H*). This expression is located precisely within a small cluster of non-proliferative dental epithelial cells (*Figure 4Ab*), marked by a lack of PCNA expression (*Figure 4Aa*). The importance of Fgf signals in regulating the differential proliferation of the EK relative to surrounding dental epithelium (*Jernvall et al., 1994*; *Kettunen et al., 2000*) together with the highly specific expression of *fgf10* within this non-proliferative region provides strong support of an EK in the catshark. The similarity in expression between marsupial mammals and sharks raises the possibility that loss of *Fgf10* in the EK in placental mammals is a lineage-specific modification in EK signalling.

During the induction of mammalian molars, Bmp4 is involved in reciprocal epithelial to mesenchymal signalling, during which the odontogenic potential shifts from the dental epithelium to the dental mesenchyme. Concordant with this shift in odontogenic potential, *bmp4* shifts from epithelial and mesenchymal expression, to mesenchymal only (*Vainio et al., 1993*). As with *fgf3* (*Figure 3G*), *lef1* and *β-catenin* (*Figure 3A, B*; *Rasch et al., 2016*), *bmp4* is also expressed in the dental epithelium and condensing mesenchyme during bud stage in the catshark (*Figure 2—figure supplement 1A*). The increase in mesenchymal *bmp4* expression between bud (*Figure 2—figure supplement 1A*) and cap stages (*Figure 2—figure supplement 1B*) reflects the epithelial to mesenchymal shift in odontogenic potential observed in mammals (*Vainio et al., 1993*). However, unlike in mammals, *bmp4* remains expressed within sub-regions of the dental epithelium throughout the duration of morphogenesis (*Figure 3I*; *Figure 2—figure supplement 1*). Given the epithelial expression of *bmp4*, during bud stage, it had previously been suggested that it may play a role in putative EK signalling in the catshark (*Rasch et al., 2016*). Following the epithelial to mesenchymal shift in Bmp4 of the mouse, it is then secondarily upregulated within the EK during late cap stage (*Jernvall et al., 1998*). However, we do not observe any secondary upregulation within the putative EK in the catshark. Instead, its expression is rapidly downregulated within the apical tip of the tooth (*Figure 3I*; *Figure 4D*). This region of bmp4 downregulation corresponds specifically to a restricted group of non-proliferative epithelial cells (*Figure 4Dc*). Throughout later stages of morphogenesis, bmp4 becomes further restricted to only the lateral epithelial cells of the developing tooth (*Figure 4E*). The precise exclusion of *bmp4* from the non-proliferative tooth apex (*Figure 4D, E*) raises the possibility that instead of regulating putative EK signalling, it is involved in the restriction of other EK markers and the signalling centre itself (*Kassai et al., 2005*). However, we cannot rule out the involvement of bmp4 in the induction of the later forming cusps of the shark tooth. Interestingly, the expression of *bmp4* in the shark tooth mirrors an equivalent expression pattern to the BMP inhibitor ectodin/sostdc1 in the inner dental

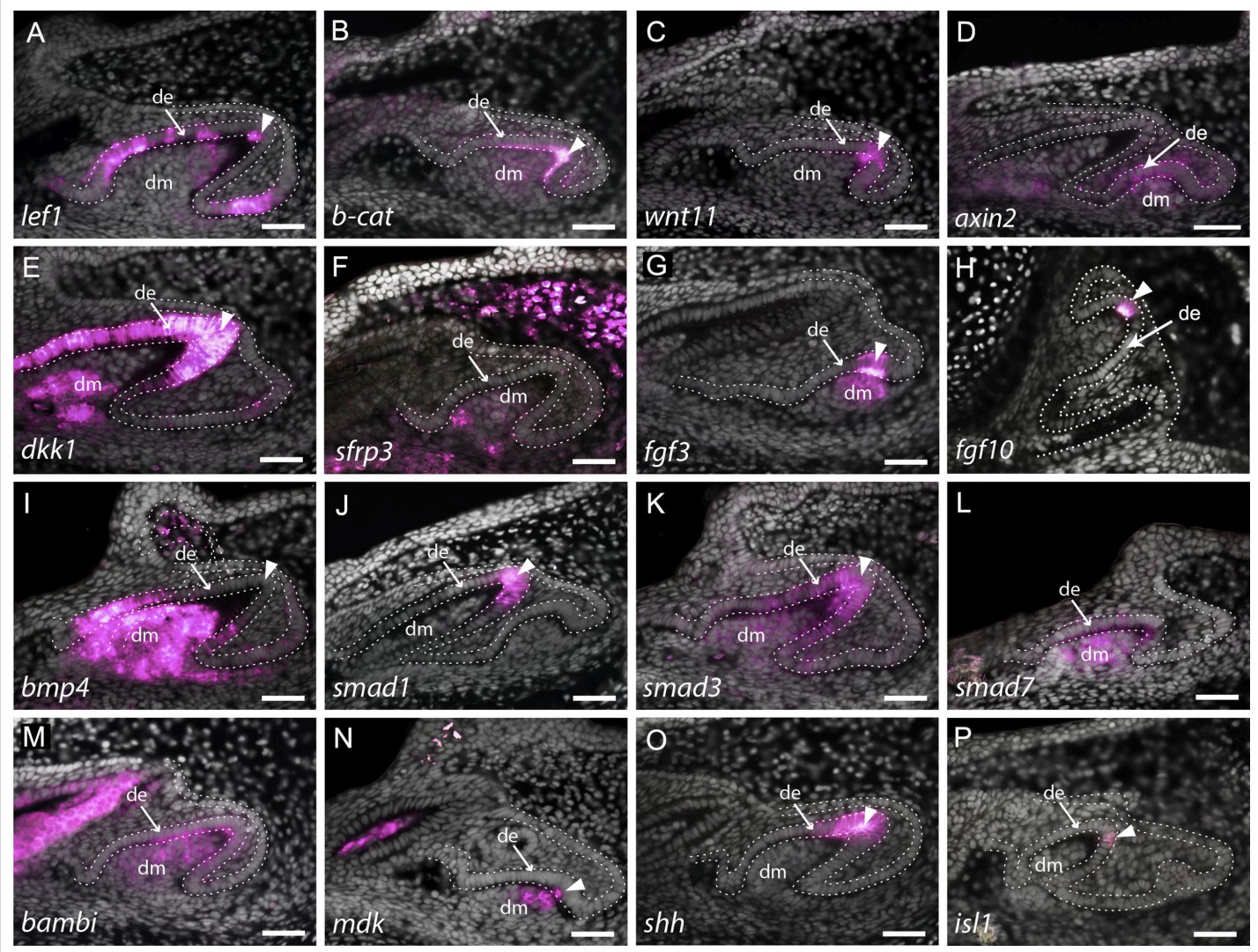

**Figure 3.** Expression of enamel knot (EK) markers during catshark dental morphogenesis. In situ hybridisation assay on sagittal paraffin sections of late stage 32 catshark jaws reveals the expression of markers involved in major developmental pathways, including canonical Wnt signalling (**A–F**), fibroblast growth factor (Fgf) signalling (**G and H**) and bone morphogenetic protein (Bmp) signalling (**I–L**). Wnt markers, *lef1* (**A**), *β*-catenin, (**B**) and *wnt11* (**C**) are all expressed within the apical tip of the developing tooth during cap stage. Axin2 is expressed in the early dental epithelium, not specifically associated with the apical tip (**D**). Weak expression can also be seen within the dental mesenchyme and surrounding dental epithelium in *lef1* (**A**) and *β*-catenin (**B**). Weak expression of the Wnt inhibitor, *sfrp3* (**F**), is found within the dental mesenchyme during cap stage, whereas *dkk1* is also found within the dental epithelium later in morphogenesis (**E**). The expression of Fgf markers *fgf3* (**G**) and *fgf10* (**H**) is highly specific to the EK during late bud to early cap stage, with *fgf3* weakly expressed in the dental mesenchyme, below the EK. *bmp4* (**I**) expression is absent from the EK during late morphogenesis, but is strongly expressed within both the rest of the dental epithelium and dental papilla. *smad1* (**J**) is found within the apical tip of the dental epithelium, whilst *smad3* (**K**) is also expressed in the dental mesenchyme during late cap stage. Bmp inhibitors *smad7* (**L**) and *bambi* (**M**) are both expressed throughout both epithelium and mesenchyme of developing teeth. *mdk* (**N**) and *isl1* (**P**) expression is restricted to a few cells of the EK, with *mdk* also expressed within the underlying dental mesenchyme. In contrast, *shh* (**O**) expression is extensive and broad, but is still restricted to the apical tip of the dental epithelium. Gene expression is false coloured in magenta. White arrowhead points to expression within the apical tip of developing teeth and putative EK. White dotted lines depict the columnar basal epithelial cells of the dental lamina and dental epithelium. DAPI nuclear stain is false coloured in grey. All images are of lower jaws, except H, which is of the upper jaw. Scale bars are 50 μm. de, dental epithelium; dm, dental mesenchyme.

epithelium (*Figure 2—figure supplement 1F*); a phenocopy of the mouse tooth expression pattern that defines the region around the ectodermal signalling centre (*Laurikkala et al., 2003*). This expression pattern supports a conserved dental epithelial signalling centre between chondrichthyan and mammalian teeth.

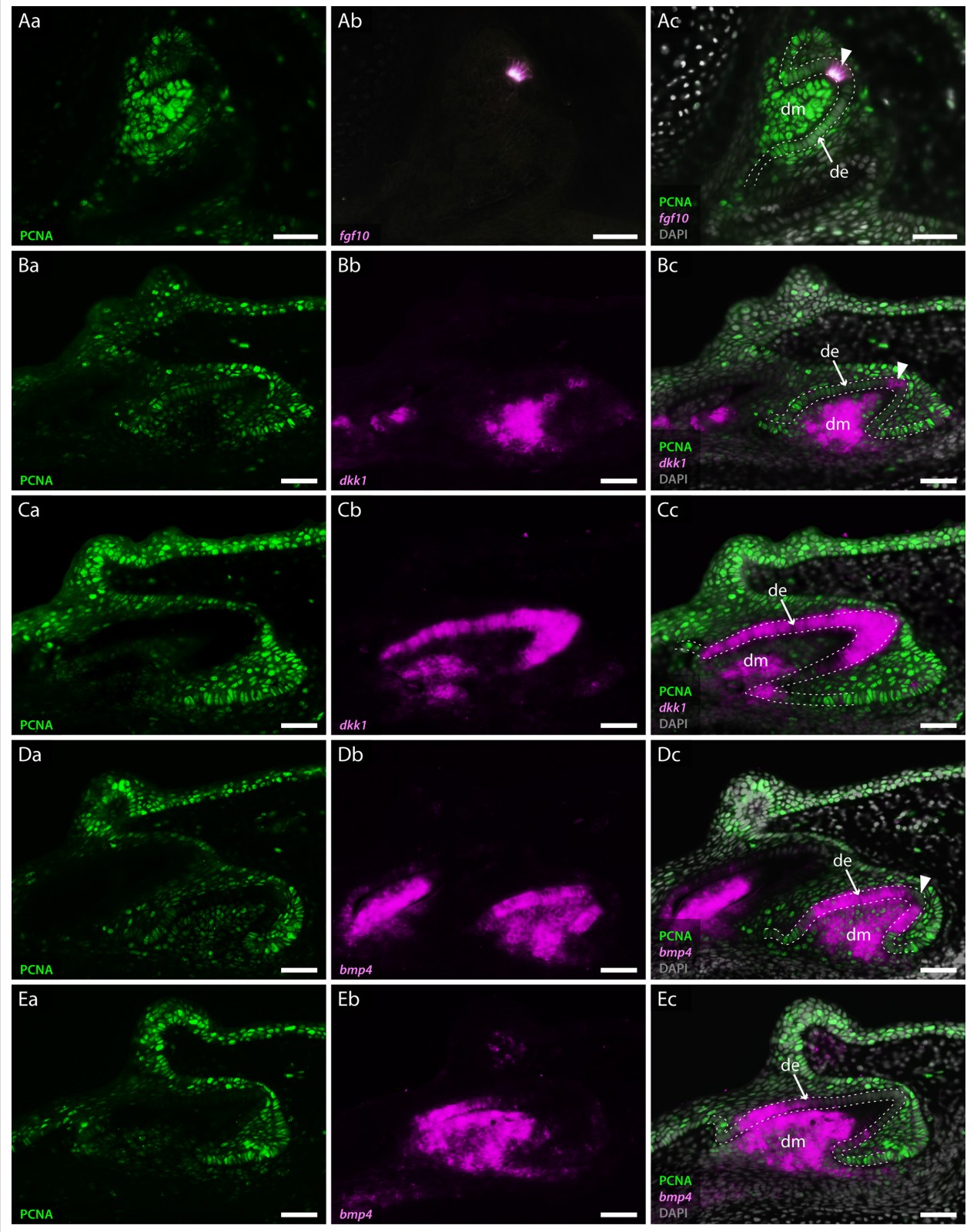

**Figure 4.** Co-expression of odontogenic markers regulating dental morphogenesis with proliferative cell nuclear antigen (PCNA). Double section in situ hybridisation/immunohistochemistry on catshark jaws reveals co-expression of markers involved in enamel knot (EK) signalling and PCNA, within teeth undergoing cap stage (**A, B, and D**) and late morphogenesis (**C and E**). Images Aa–Ea reveal expression of PCNA. Images Ab–Eb reveal expression of in situ hybridisation markers. Images Ac–Ec reveal co-expression of PCNA and in situ hybridisation markers. PCNA expression is absent

*Figure 4 continued on next page*

*Figure 4 continued*

from the tip of the dental epithelium, corresponding to the EK in cap stage teeth (Aa, Ba, and Da). The extent of PCNA expression within developing teeth decreases as teeth undergo morphogenesis (Ca and Ea). *fgf10* is expressed within a small subset of dental epithelial cells corresponding to the EK (Ab). Its expression is inversely complementary to the expression of PCNA (Ac). *dkk1* expression is initially upregulated in the dental mesenchyme during cap stage (Bb), with restricted epithelial expression present only within non-proliferative cells of the EK (Bc). During late morphogenesis *dkk1* expression weakens within the dental mesenchyme, with observable upregulation of its expression throughout the entire dental epithelium (Cb and Cc). Unlike *dkk1* and *fgf10*, *bmp4* is absent from the apical tip of the dental epithelium throughout dental morphogenesis (Db and Eb). *fgf10* (**A**) is shown in the upper jaw; *dkk1* (**B and C**) and bmp4 (**D and E**) are shown in the lower jaw. Gene expression is shown in magenta, PCNA protein expression is in green, and DAPI nuclear stain in grey. White dotted lines depict the outer dental epithelium of the developing tooth. Scale bars are 50 µm. de, dental epithelium; dm, dental mesenchyme.

Midkine (mdk) has previously been observed within the apical tip of developing teeth in the catshark (*S. canicula*) and little skate (*Leucoraja erinacea*), corresponding to the putative EK-like signalling centre (*Rasch et al., 2020*; *Rasch et al., 2016*). We find *mdk* with an overlapping expression pattern to *fgf3* in both the apical epithelial cells and the underlying mesenchymal papilla (*Figure 3N*), whilst shh is found early in tooth bud formation, exclusively within the apical tip of the dental epithelium (*Figure 3O*; *Figure 2G*) before its expansion throughout the inner dental epithelium (*Figure 2H, I*). Another marker with a similar and restricted expression pattern to *fgf10* is *isl1* (*Figure 3P*). isl1 expression is only observed in a few apical cells associated with the putative primary EK.

Canonical Wnt signalling plays a crucial role in the formation of dental cusps in mammals (*Kratochwil et al., 2002*; *Liu et al., 2008*). In the catshark, the transcription factor *lymphoid enhancer binding factor-1* (*lef1*) (*Figure 3A*) and the transcriptional co-factor *β-catenin* (*Figure 3B*) are both expressed within developing bell stage teeth. Weak expression can be seen in the dental mesenchyme (*Figure 3*: dm) and regions of the dental epithelium (*Figure 3*: de). However, there is also an observable upregulation of their expression within the apical dental epithelium (*Figure 3*: white arrowhead). Wnt11 has been previously identified as an activator of non-canonical Wnt signalling, although its expression has also been shown to stimulate canonical Wnt signalling in a case-specific context (*Liu et al., 2008*). We note the upregulation of *wnt11* specifically within the apical dental epithelium during cap stage (*Figure 3C*). The downstream target of canonical Wnt signalling, *axin2*, is also expressed within the early dental epithelium; however, its expression is not specifically restricted to the apex (*Figure 3D*).

Canonical Wnt antagonists *Dickkopf 1* (*dkk1*) and *frizzled-related protein 3* (*sfrp3*) are also expressed within the developing tooth (*Figure 3E, F*, respectively). *dkk1* is seen within both the dental papilla and throughout the dental epithelium in late morphogenesis (*Figures 3E and 4C*). Reduced proliferation is a key requirement of an EK, with differential proliferation rates within the dental epithelium leading to the formation of dental cusps (*Jernvall et al., 1994*). Double in situ/ immunohistochemistry for *dkk1* and proliferative cell nuclear antigen (PCNA) revealed a small number of non-proliferative cells at the tip of the tooth during cap stage (*Figure 4Ba*), with a marked reduction in proliferation throughout the entire dental epithelium late morphogenesis (*Figure 4Ca*). Although *dkk1* is expressed throughout the entire dental epithelium during late morphogenesis (*Figure 3E*), during cap stage its epithelial expression is specifically restricted to the non-proliferative cells corresponding to the putative EK (*Figure 4B*). In the pig (*Wu et al., 2017*), *dkk1* expression is restricted to the dental mesenchyme, with no expression observed within the dental epithelium. In contrast to *dkk1*, *sfrp3* is specifically restricted to the dental mesenchyme, with weak expression within the dental papilla (*Figure 3F*). In the mouse, *sfrp3* is also restricted to the mesenchyme (*Sarkar and Sharpe, 1999*), though its expression is much stronger than that observed in the shark (*Figure 3F*). Not all canonical Wnt-related markers are expressed in equivalent tissue types during the cap to early bell stage of dental development between mammals and the shark. However, there is expression of Wnt markers within both the dental epithelium and dental mesenchyme, with an isolated yet upregulated region of expression within the putative EK. The expression of canonical Wnt antagonists *dkk1* and *sfrp3* may also be playing a role in restricting Wnt activity within the dental mesenchyme.

The Smad protein family plays a crucial role in TGFβ signal transduction, including Bmp signalling (*Massagué, 2012*). Here, we show the expression of *smad1*, *smad3*, and *smad7* during dental morphogenesis (*Figure 3J, K and L*). Smad1 and Smad3 are phosphorylated following Bmp signalling and enable its signal transduction, whilst Smad7 is a negative feedback regulator of TGFβ signalling (*Massagué, 2012*; *Xu et al., 2003*). *smad1* (*Figure 3J*) is expressed within the putative EK at the

tip of the developing tooth, whilst *smad3* (*Figure 3K*) and *smad7* (*Figure 3L*) are both expressed throughout the dental epithelium and dental mesenchyme. Furthermore, as with *smad7*, BMP and activin membrane-bound inhibitor (*bambi*) is expressed in both dental epithelium and dental papilla (*Figure 3M*). Although interactions between these signalling molecules are complex, the expression of Bmp markers within and around the putative EK indicates a role for Bmp in EK signalling.

EK-like units comprise a small number of cells at the very apical tip of a developing tooth and can be hard to morphologically distinguish from surrounding dental epithelium in sagittal cross sections. Therefore, we also used whole mount in situ hybridisation to examine the expression of markers throughout the entire tooth unit.

## Whole mount gene expression patterns in first-generation shark teeth

As the first dental generation develops superficially, prior to full invagination of the dental lamina, teeth can be seen developing directly on the oral surface. We carried out whole mount in situ hybridisation experiments for *fgf3*, *fgf10*, *bmp4*, and *shh* on early stage 32 (~90 dpf) catshark embryos, during which the first dental generation undergoes morphogenesis (*Figure 5*). This allowed us to identify whether markers expressed within the apical tip of developing teeth in sagittal sections were specifically restricted within cusp forming regions, in whole mount.

As with our section in situ hybridisation data, *bmp4* is excluded from the epithelium at the leading edge of the tooth, in whole mount. Instead, its expression appears to be primarily restricted to the dental papilla in both upper (*Figure 5A*) and lower jaws (*Figure 5B*). In contrast, *shh* expression can be seen within the dental epithelium and specifically upregulated within the leading edge of the tooth (*Figure 5E and F*; *Figure 2*). *shh* is confined to the EK in mammalian molars (*Hardcastle et al., 1998*; *Vaahtokari et al., 1996*), before spreading throughout the inner enamel epithelium later during tooth morphogenesis (*Gritli-Linde et al., 2002*; *Kavanagh et al., 2007*), and interestingly we see strong complementary expression within both the primary cusp (*Figure 5E and F*: black arrowhead) and later morphogenesis in the shark cusps (*Figure 5E and F*: white arrowhead; *Figure 2*). Depending on the stage of morphogenesis, the extent of *shh* expression within the inter-cusp dental epithelium is variable (see *Figure 2*). Alongside playing a putative role in EK signalling, the expression of *shh* within the epithelium at the leading edge of the tooth suggests that it may also be involved in establishing an anterior to posterior growth gradient within the tooth unit. In whole mount, we observe an additional gradient of tooth stages that are reflected in the variation in observed *shh* expression, with more posterior tooth positions along the jaw arc at earlier stages of morphogenesis (*Figure 5E and F*). More anterior teeth show apical expression of *shh* in the secondary cusps, whereas the more posterior tooth positions have *shh* expression associated with initial tooth buds, primary cusp morphogenesis, and even the inter-tooth epithelium, where new tooth positions are being initiated (*Figure 5E" and F"*).

Unlike with *shh*, *fgf3* and *fgf10* expression is clearly associated with both the primary cusp (*Figure 5C, D and G*: black arrowhead) and secondary cusps (*Figure 5C, D and G*: white arrowhead). Whilst there is expression of *fgf3* within the dental mesenchyme, its epithelial expression is restricted solely to the site of future cusps. *fgf10* is exclusively expressed within the cusp forming dental epithelium, in agreement with the expression pattern observed in section (*Figure 4A*). The expression patterns of *shh*, *bmp4*, and *dkk1*, in or around the dental cusps, and more importantly the precise restriction of *fgf3* and *fgf10* to these sites (*Figure 5*), demonstrate the conservation of an EK-like signalling centre at the dental cusp sites in the shark.

## Wnt signal perturbation and geometric morphometric analysis of shark tooth morphogenesis

Wnt/β-catenin signalling has been implicated upstream of key signalling pathways (Bmp, Fgf) as well as in the regulation of transcription factors (e.g., Msx genes) regulating during dental morphogenesis (*Kratochwil et al., 2002*; *Liu et al., 2008*). Having identified the presence of a putative EK during dental development in the catshark, we wanted to test the role of canonical Wnt signalling in the regulation of dental shape. IWR-1-endo is a canonical Wnt antagonist. It functions through stabilising cytoplasmic Axin2, in turn increasing proteasome-mediated degradation of cytoplasmic β-catenin, leading to a decrease in canonical Wnt signalling (*Chen et al., 2009*). In contrast, CHIR99021 selectively inhibits the kinase activity of GSK*3* (*Ring et al., 2003*). This prevents the formation of the β-catenin

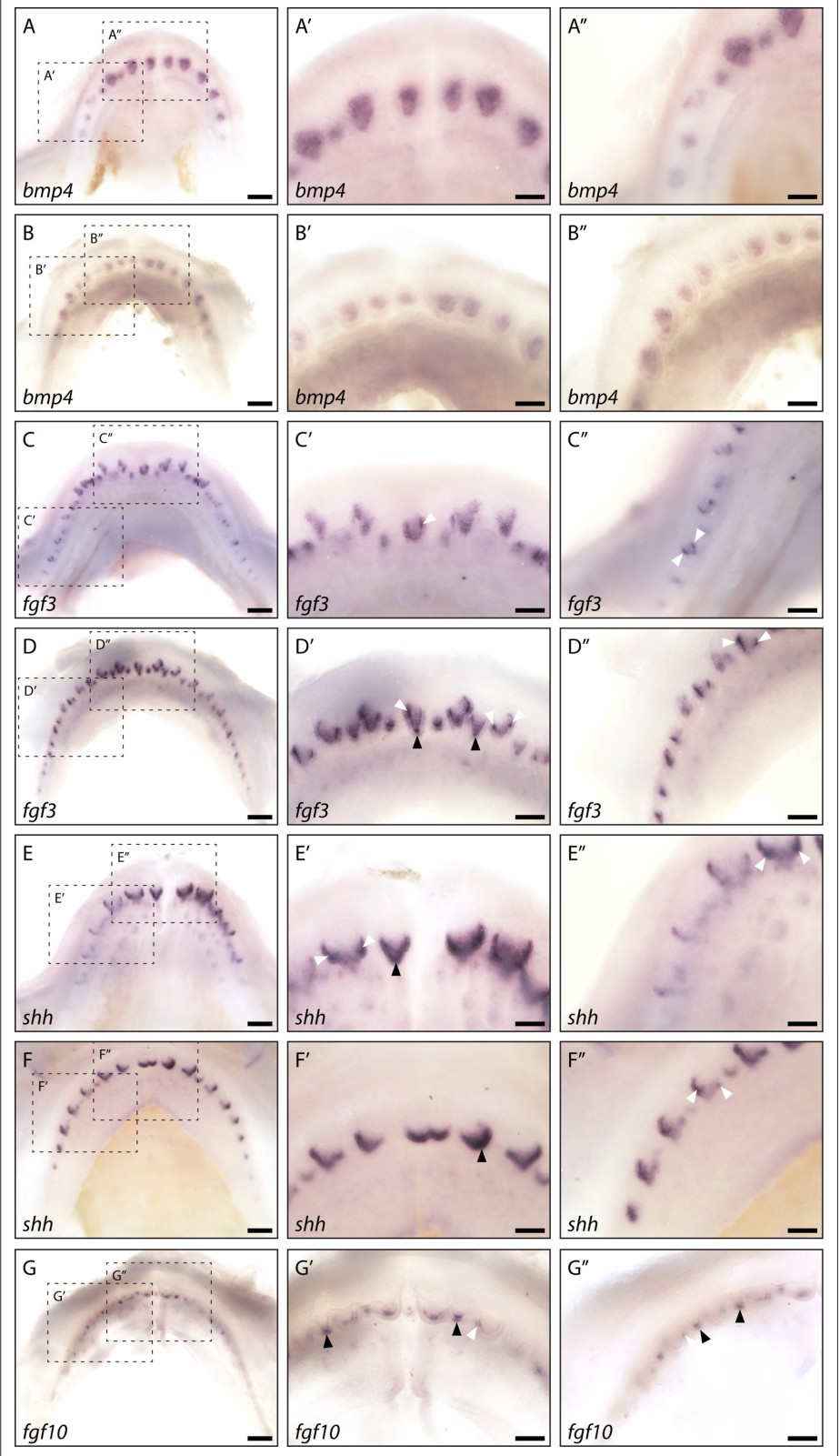

**Figure 5.** Whole mount in situ hybridisation reveals enamel knot (EK)-specific gene expression. Whole mount in situ hybridisation on catshark lower (**A, C, and E**) and upper (**B, D, F, and G**) jaws highlights the expression of *bmp4* (**A and B**), *fgf3* (**C and D**), *shh* (**E and F**), and *fgf10* (**G**) across the jaw. Images are of early stage 32 (~90 days post fertilisation [dpf]) samples, with developing first-generation teeth visible on the jaw margin. Images A–G are

*Figure 5 continued on next page*

*Figure 5 continued*

low magnification images. Images A'–G' are magnified images of the central teeth along the jaw. Images A"–G" are magnified images of the left lateral side of the jaw. *bmp4* expression is visible throughout the dental papilla, but appears absent from the dental epithelium (**A', A", B', and B"**). *fgf3* expression is strongly upregulated within both primary EKs (D': black arrowheads) and secondary EKs (C', C", D', and D": white arrowheads). Weaker *fgf3* expression is also noted within the dental mesenchyme (**C', C", D', and D"**). *shh* expression is absent from the dental mesenchyme. However, its expression can be seen throughout the dental epithelium at the leading edge of the developing tooth (**E', E", F', and F"**). Although its expression is not restricted to the EK throughout morphogenesis, clear *shh* expression present within both the primary EKs (E' and F': black arrowheads) and secondary EKs (E', E", and F": white arrowheads). As with *fgf3*, *fgf10* expression is also restricted to the future cusp forming primary (G' and G": black arrowheads) and secondary EKs (G' and G": white arrowhead), however its expression does not extend into the dental mesenchyme. Dotted black boxes in A–G depict a magnified region in A'–G' and A"–G". Scale bars are 250 μm in A–G and 125 μm in A'–G' and A"–G".

destruction complex and in turn leads to a stabilisation of cytoplasmic β-catenin and a subsequent increase in canonical Wnt signalling (*Wagman et al., 2004*). Wnt signalling was both upregulated and downregulated using 2 μM CHIR99021 and 1 μM IWR-1-endo, respectively. We treated samples for 14 days, allowing for the initiation of, on average, one tooth generation. Samples were treated at approximately 100 dpf (mid-stage 32), by which point, one to two generations are undergoing morphogenesis.

-catenin signalling has been implicated upstream of key signalling pathways (Bmp, Fgf) as well as in the regulation of transcription factors (e.g., Msx genes) regulating during dental morphogenesis (*Kratochwil et al., 2002*; *Liu et al., 2008*). Having identified the presence of a putative EK during dental development in the catshark, we wanted to test the role of canonical Wnt signalling in the regulation of dental shape. IWR-1-endo is a canonical Wnt antagonist. It functions through stabilising cytoplasmic Axin2, in turn increasing proteasome-mediated degradation of cytoplasmic β-catenin, leading to a decrease in canonical Wnt signalling (*Chen et al., 2009*). In contrast, CHIR99021 selectively inhibits the kinase activity of GSK*3* (*Ring et al., 2003*). This prevents the formation of the β-catenin destruction complex and in turn leads to a stabilisation of cytoplasmic β-catenin and a subsequent increase in canonical Wnt signalling (*Wagman et al., 2004*). Wnt signalling was both upregulated and downregulated using 2 μM CHIR99021 and 1 μM IWR-1-endo, respectively. We treated samples for 14 days, allowing for the initiation of, on average, one tooth generation. Samples were treated at approximately 100 dpf (mid-stage 32), by which point, one to two generations are undergoing morphogenesis.

Following treatment, samples were left to recover for a further 28 days before we examined the treatment effect upon final tooth shape. Drastic shifts in the shape of teeth were observed following treatment (*Figure 6*). Abnormalities in cusp development were seen across both IWR-1-endo and CHIR99021-treated samples (*Figure 6*; *Figure 6—figure supplement 1*; and *Figure 7*), however, given the presence of reduced cusps and uneven tooth edges (serrated in appearance), quantifying the number of cusps was not feasible. To accurately compare changes in tooth shape following treatment, we carried out two-dimensional (2D) geometric morphometric measurements of mineralised teeth following treatment (*Figure 6C*).

Landmark-based geometric morphometrics uses Cartesian coordinates instead of traditional linear measurements in order to accurately measure biological shape (*Seetah et al., 2014*). A measure of size known as the centroid size can then be obtained from these coordinates. This is calculated as the square root of the sum of squared distances from each landmark to the centroid (*Klingenberg, 2016*). The coordinates obtained from the landmarks are then transposed, scaled (relative to the centroid size), and rotated, giving a final measurement of shape which can be compared across different treatments (*Klingenberg, 2016*). For our analysis, we used a 2D landmark-based geometric morphometric approach. We placed two fixed homologous landmarks on each tooth, one at the tip of the primary cusp and one at the base of the tooth. Eighteen sliding semi-landmarks were then distributed evenly between these two points on either side of the tooth, given that there were no other homologous features shared by all samples (*Figure 6B*). Sliding semi-landmarks are allowed to move between one another so that they best match corresponding points between other samples. The underlying assumption is that the curves along which the landmarks slide are homologous, even if the landmarks themselves are not (*Perez et al., 2006*).

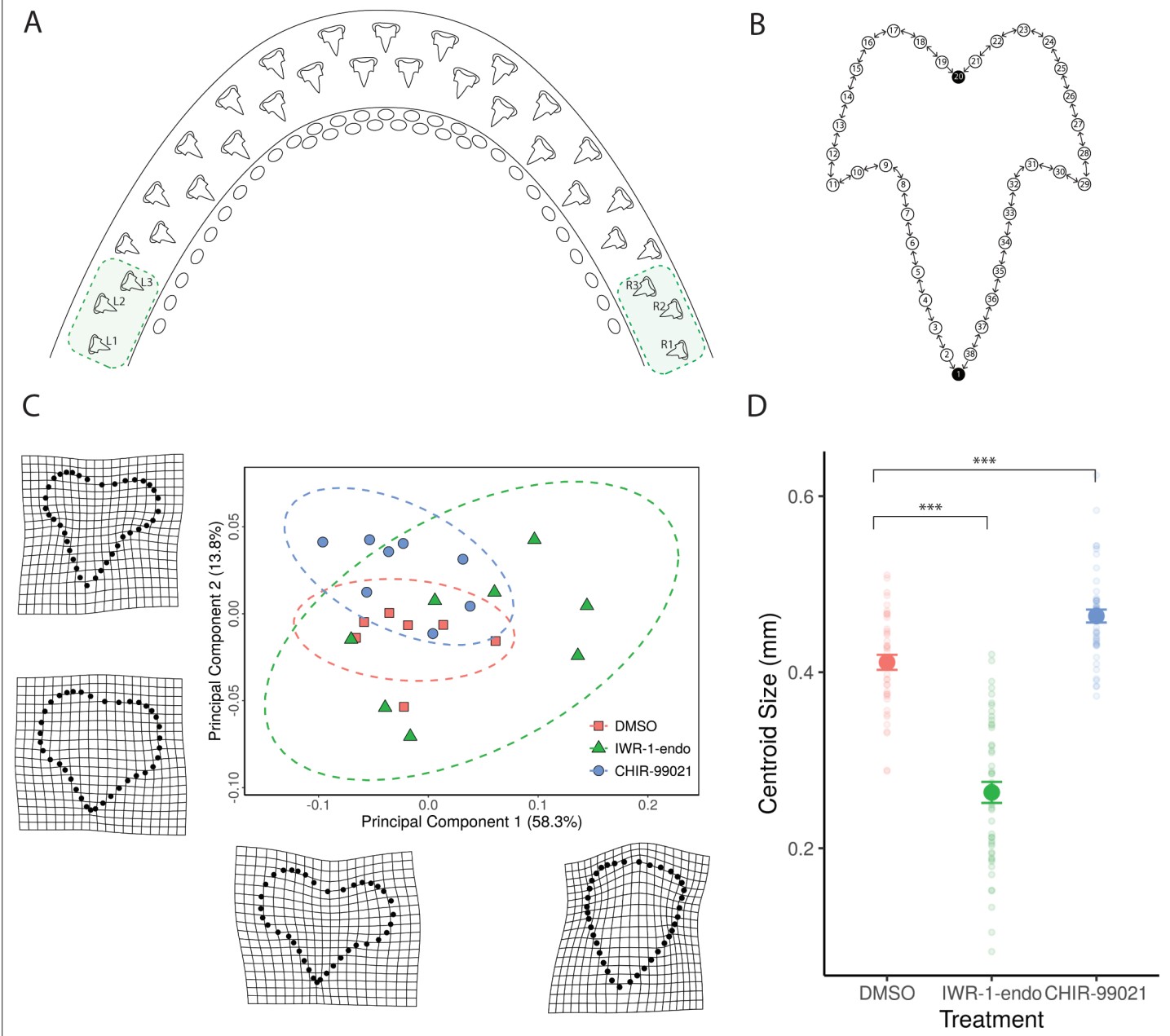

**Figure 6.** Geometric morphometric analysis reveals change in shape and size following canonical Wnt manipulation. Small molecule treatments consisted of a 2-week treatment with 0.1% DMSO (control), 1 µM IWR-1-endo, and 2 µM CHIR99021 and a subsequent 4-week recovery. The three most lateral teeth at both left and right lower jaw margins (**A**) were included in the geometric morphometric analysis. A total of 38 landmarks were used to label tooth shape (**B**). Two fixed landmarks were placed at the tip of the primary cusp and at the base of the tooth, represented by black circles. Eighteen sliding semi-landmarks were placed on either side of the tooth (white circles), which were allowed to move relative to adjacent landmarks. The direction of movement is depicted by directional arrows in the schematic (**B**). Following Procrustes alignment of landmark coordinates and principal component analysis of the resulting shapes, the average PC1 and PC2 scores for each sample were plotted in order to depict the position of treated samples within a given shape space (**C**). PC1 accounted for 58.3% of the variation observed between samples, whereas PC2 accounted for 13.8%. Warpgrids shown in C reveal representative shapes at maximum and minimum PC1 and PC2 values. There is a significant effect of treatment on overall tooth shape (Procrustes ANOVA: $R^2=0.09174$, $F_{2,115}=8.2771$, $p<0.001$), whilst controlling for variation within samples. There is also a significant effect of sample on shape (Procrustes ANOVA: $R^2=0.27092$, $F_{20,115}=2.4442$, $p<0.001$). Aside from shape, centroid size was also measured following treatment (**D**). There is a significant effect of treatment on centroid size (ANOVA: $F_{2,115}=235.5886$, $p<0.001$), whilst controlling for variation within samples. There is also a significant effect of sample on centroid size (ANOVA: $F_{20,115}=7.2957$, $p<0.001$). Faint circular points are plots of each individual data point, revealing the distribution of the data. Error bars represent standard error.

The online version of this article includes the following source data and figure supplement(s) for figure 6:

*Figure 6 continued on next page*

*Figure 6 continued*

**Source data 1.** Principal component analysis (PCA) data.

**Source data 2.** Procrustes shape data.

**Source data 3.** Principal component analysis (PCA) landmarks/scale.

**Source data 4.** Csize by treatment.

**Source data 5.** GMM metadata.

**Source data 6.** CurveSlide.

**Figure supplement 1.** Representative experimental images from geometric morphometric principal component analysis.

**Figure supplement 2.** Canonical Wnt signalling control experiments: cleared and stained (Alizarin Red S) lower jaws (dorsal views) of *Scyliorhinus canicula*, DMSO controls.

**Figure supplement 3.** Canonical Wnt signalling inhibition experiments: cleared and stained (Alizarin Red S) lower jaws (dorsal views) of *Scyliorhinus canicula*, treated with 1 μM IWR-1-endo (Wnt pathway inhibitor).

**Figure supplement 4.** Canonical Wnt signalling activation experiments: cleared and stained (Alizarin Red S) lower jaws (dorsal views) of *Scyliorhinus canicula*, treated with 2 μM CHIR-99021 (small molecule Wnt pathway activator; GSK3 inhibitor).

**Figure supplement 5.** Scatter plot illustrating all individual tooth data points from principal component analysis illustrated in *Figure 6C*.

We chose to compare the shape of the three most lateral teeth of both the left and right lower jaw margins (*Figure 6A*) to mitigate variation observed between dental generations elsewhere in the jaw. Furthermore, there is substantial variation in tooth number during this stage of development and therefore a direct comparison of tooth positions elsewhere in the jaw is problematic. Dental differentiation and mineralisation also take place unidirectionally, from the apex towards the base of the tooth. This wave of mineralisation could lead to further variability in measurements of final tooth shape. After carrying out a general Procrustes analysis (GPA), we found a significant effect of treatment on overall tooth shape (Procrustes ANOVA: $R2=0.09174$, $F2,115=8.2771$, $p<0.001$), whilst controlling for variation within samples. We also found a significant effect of sample on shape (Procrustes ANOVA: $R2=0.27092$, $F20,115=2.4442$, $p<0.001$), highlighting variation found within sample measurements.

To look at the variation in tooth shape as a result of treatment, we averaged the six teeth measured per sample and carried out principal component analysis (PCA). PCA revealed that 58.3% of the variation in tooth shape observed between samples can be explained by a single axis of variation (PC1), with a further 13.8% explained by a second axis of variation (PC2) (*Figure 6C*). As a result of high variation in dental shape, secondary cusps are not fully represented in the morphometric analysis. However, their presence can somewhat be revealed through a bulging of the tooth at sites adjacent to the primary cusp (*Figure 6C*: warpgrids). The shift in dental shape observed across PC1 attributes to a change in width of the tooth and an apparent change in cusp number. Four out of eight, IWR-1-endo-treated samples can be seen exhibiting extreme unicuspid dental morphologies (PC1=0.05–0.15), whilst five out of eight CHIR99021-treated samples exhibit wide teeth with more distinctive cusps (PC1=–0.1 to 0).

When looking at the variation in tooth shapes between individual teeth (not averaged by sample), we observe a significant increase in the variation of dental shapes across the PC1 axis (38.2% of overall variation) following Wnt downregulation with IWR-1-endo (pairwise F-test: $F41,47=0.277$, $p<0.001$) (*Figure 6—figure supplement 5*). These results show that dental shape is being affected as a result of canonical Wnt manipulation, although it is difficult to discern directional changes in morphology. We observe not only shifts in shape as a result of treatment, but also substantial changes in the size of teeth. Centroid size derived from GPA is a measure of an object's scale. This provides more details than directional measurements such as length or area which are affected by an object's shape (*Seetah et al., 2014*; *Klingenberg, 2016*). Following treatment, we found a significant effect of treatment on centroid size (ANOVA: $F2,115=235.5886$, $p<0.001$), whilst controlling for variation within samples (*Figure 6D*). However, there was also a significant effect of the sample on centroid size (ANOVA: $F20,115=7.2957$, $p<0.001$).

In order to compare the mean centroid size between treatments, we carried out post hoc Tukey multiple comparisons of means test. There is a significant decrease in centroid size (–35.9%) following IWR-1-endo treatment (Tukey: $p<0.001$), whilst conversely there is a significant increase in centroid size (+12.8%) as a result of CHIR99021 treatment relative to control samples (Tukey: $p<0.001$)

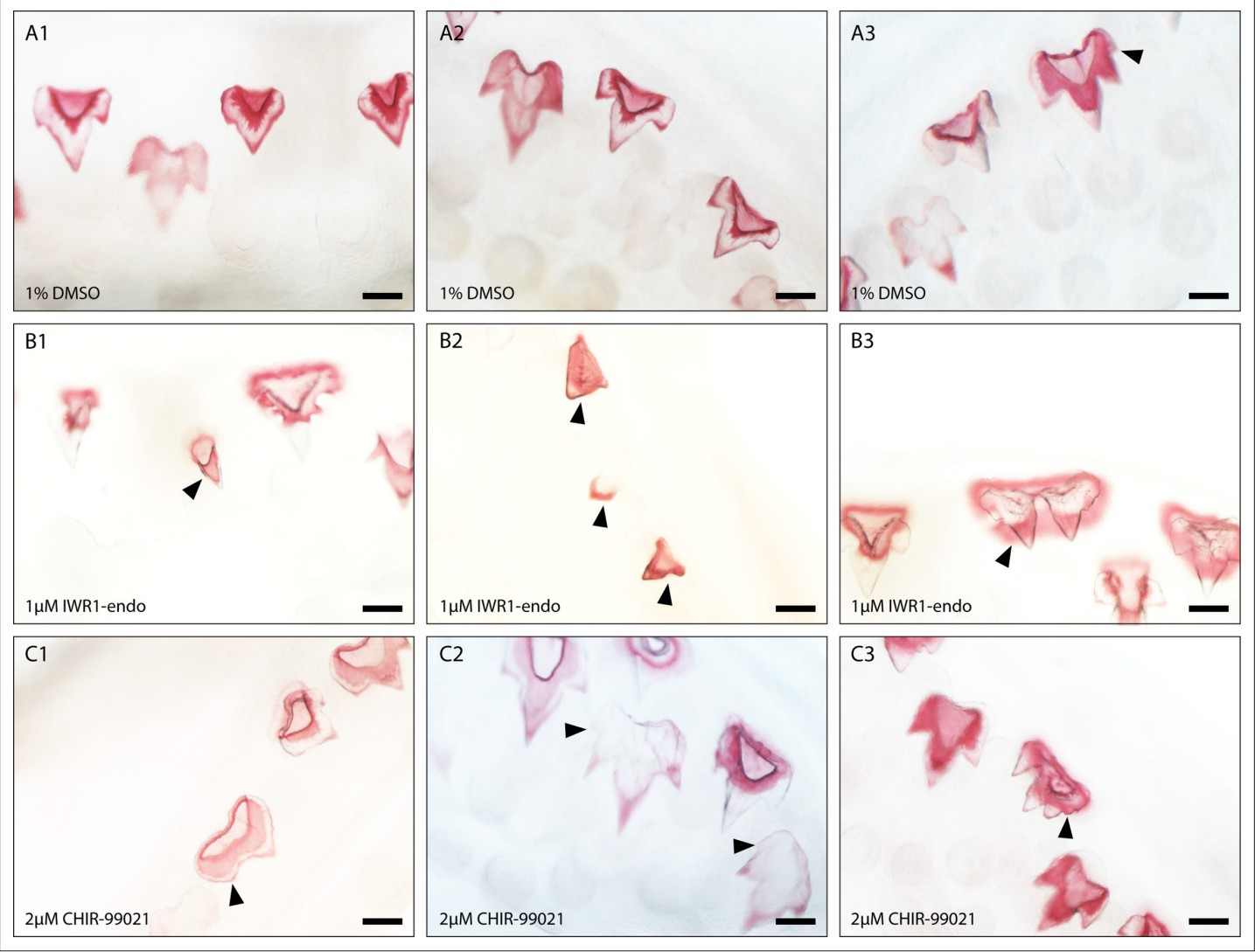

**Figure 7.** Dental diversity following canonical Wnt manipulation. Selected images depicting dental diversity following 0.1% DMSO (control) (**A1–A3**), 1 µM IWR-1-endo (**B1–B3**), and 2 µM CHIR9902 (**C1–C3**) treated lower jaws following 2-week treatment and 4-week recovery. 0.1% DMSO are relatively similar in shape, although the presence of 4 cusps is observed in a small subset of teeth (**A3**). Following 1 µM IWR-1-endo treatment, there are numerous mineralised unicuspid and stunted teeth (**B1 and B2**). There are also teeth which appear duplicated in nature, but which are connected to a single root (**B3**). In contrast, 2 µM CHIR9902 treatment leads to a widening of the teeth and defects in the position of cusps (**C1**) and the development of supernumerary cusps (**C2 and C3**). Black arrowheads point to cusp defects and/or shifts from the typical tricuspid dental morphology. Scale bars are 100 µm.

(*Figure 6D*). These findings reveal a directional change in tooth shape as a result of canonical Wnt signalling manipulation, given that IWR-1-endo and CHIR99021 downregulate and upregulate Wnt signalling, respectively.

## Canonical Wnt signalling during development of dental cusps

Representative images of lateral teeth included in the geometric morphometric analysis reveal clear changes in tooth size (*Figure 6—figure supplement 1*). IWR-1-endo treatment resulted in stunted teeth, with the developed teeth failing to undergo normal morphogenesis, yet successfully undergoing mineralisation (*Figure 6—figure supplement 1C*). Most lateral teeth exhibit a reduction in secondary cusp size, with some teeth exhibiting a unicuspid morphology with a complete loss of secondary cusps. Observations of dental morphology at other sites along the jaw reveal a range of dental defects following treatment (*Figure 7*). There is a consistent loss/reduction in secondary cusps along the jaw (*Figure 7B1 and B2*), which is concurrent with the morphologies observed in the lateral

teeth included in the geometric morphometric analysis, although not all teeth exhibit this phenotype (*Figure 7B3*). We also observed the development of duplicated teeth (*Figure 7B3*). This may arise from fusion of tooth buds belonging to adjacent tooth families, as a result of a loss in the zone of inhibition between tooth sites. However, given that adjacent teeth are staggered in the timing of their development, it is more likely that shifts in signalling during early morphogenesis has resulted in defects in the folding of the dental epithelium resulting in the formation of two primary cusps.

In contrast, upregulation of canonical Wnt signalling via CHIR99021 treatment resulted in teeth more similar in appearance to the standard catshark dentition. Lateral CHIR99021 teeth (*Figure 6—figure supplement 1D* and *Figure 6—figure supplement 4*) are clearly larger and wider than controls (*Figure 6—figure supplement 1B* and *Figure 6—figure supplement 2*), with regions of the teeth appearing to initiate the formation of ectopic cusps, although these do not always fully form (*Figure 6—figure supplement 1D*b: black arrowheads). Interestingly, within other regions of the jaw, we do see the development of fully formed fourth cusps (*Figure 7C2and C3*). This is similar to the phenotypes we see in juvenile catsharks, which develop teeth with up to 7 cusps. The development of fourth cusps can also be seen within some of the control teeth (*Figure 7A3*). However, the proportion of teeth exhibiting fourth cusps is significantly lower in control samples (1%), than CHIR99021-treated samples (12%) (chi-square: $\chi$-squared=16.887, df = 1, p-value < 0.001). Furthermore, the position and size of cusps in DMSO (*Figure 6—figure supplement 2*) treated samples is very consistent. There is a clear single primary cusp, with two secondary cusps equal in size. Extra cusps develop laterally and are smaller in size than the initial secondary cusps. This is not the case in CHIR99021-treated specimens. We note cases whereby secondary cusps are enlarged and can be difficult to distinguish apart from the primary cusp. We also observe teeth bearing multiple small cusps which resemble serrated teeth. The defects observed in final tooth shape highlight an important role for Wnt during dental morphogenesis. The specific directional shifts in the number of cusps associated with down- and upregulation through IWR-1-endo (*Figure 6—figure supplement 3*) and CHIR99021 treatment (*Figure 6—figure supplement 4*), respectively, implicate canonical Wnt in the regulation of shark cusp morphogenesis.

## In silico modelling of wild-type tooth shape

*Salazar-Ciudad and Jernvall, 2010*, developed a computational model (ToothMaker) capable of modelling vertebrate dentitions based on 12 cellular and 14 gene-network parameters with predetermined values. To establish the baseline parameters capable of generating a wild-type catshark tooth, we started with parameters established in modelling the seal dentition (*Salazar-Ciudad and Jernvall, 2010*). We iterated through each parameter at 10% intervals, using a process of elimination to refine the final tooth parameters (*Figure 8D*; *Supplementary file 1*: seal). A shift in cusp number is observed throughout ontogeny, with an increase from 3 cusps in the embryo, to 5 or 6 cusps in juveniles (*Figure 8A–C*). When attempting to simulate this increase in cusp number, we observed that changing the width of the initial tooth site (Bwi) together with activator-autoactivation (Act) is capable of recreating 5 and 6 cusped teeth (*Figure 8E–F*). As sharks grow, the size of new dental generations increases (*Figure 8A–C*). It is therefore possible that during successive rounds of dental regeneration, the size of the initial tooth site increases, in turn generating larger teeth with more numerous cusps.

Despite successfully modelling an increase in cusp number using these baseline seal parameters, the modelled teeth did not exhibit a pointed shark-like dental morphology. After correspondence with the software developers (*Savriama et al., 2018*), we developed a modified set of baseline parameters (*Supplementary file 1*: shark) which generated a tricuspid tooth closely resembling the embryonic shark dental morphology. When attempting to replicate the normal shark tooth development in ontogeny from tricuspid through to the 6-cusp phenotype, we found that there were two parameter modules which influenced the final shape (*Figure 8H–I*; *Supplementary file 1*). The first module concerns the balance of the activatory and inhibitory propensity of the gene network. Starting from a tricuspid baseline (*Figure 8G*), a 10% increase in activator autoactivation together with an increase in protein degradation results in a 5-cusp tooth (*Figure 8H*). Both the adjustment to protein degradation and activator/inhibitor levels needs to be met to move toward the 6-cusp phenotype (*Figure 8I*), and a concordant increase in inhibitor diffusion. Also required are adjustments to the second module of parameters, which concerns physical characteristics. An increase to the original border width of the tooth site and the border growth rate is necessary to accommodate the larger tooth (*Figure 8I*, *Supplementary file 1*). In *S. canicula*, tooth buds are closely packed and during development form an

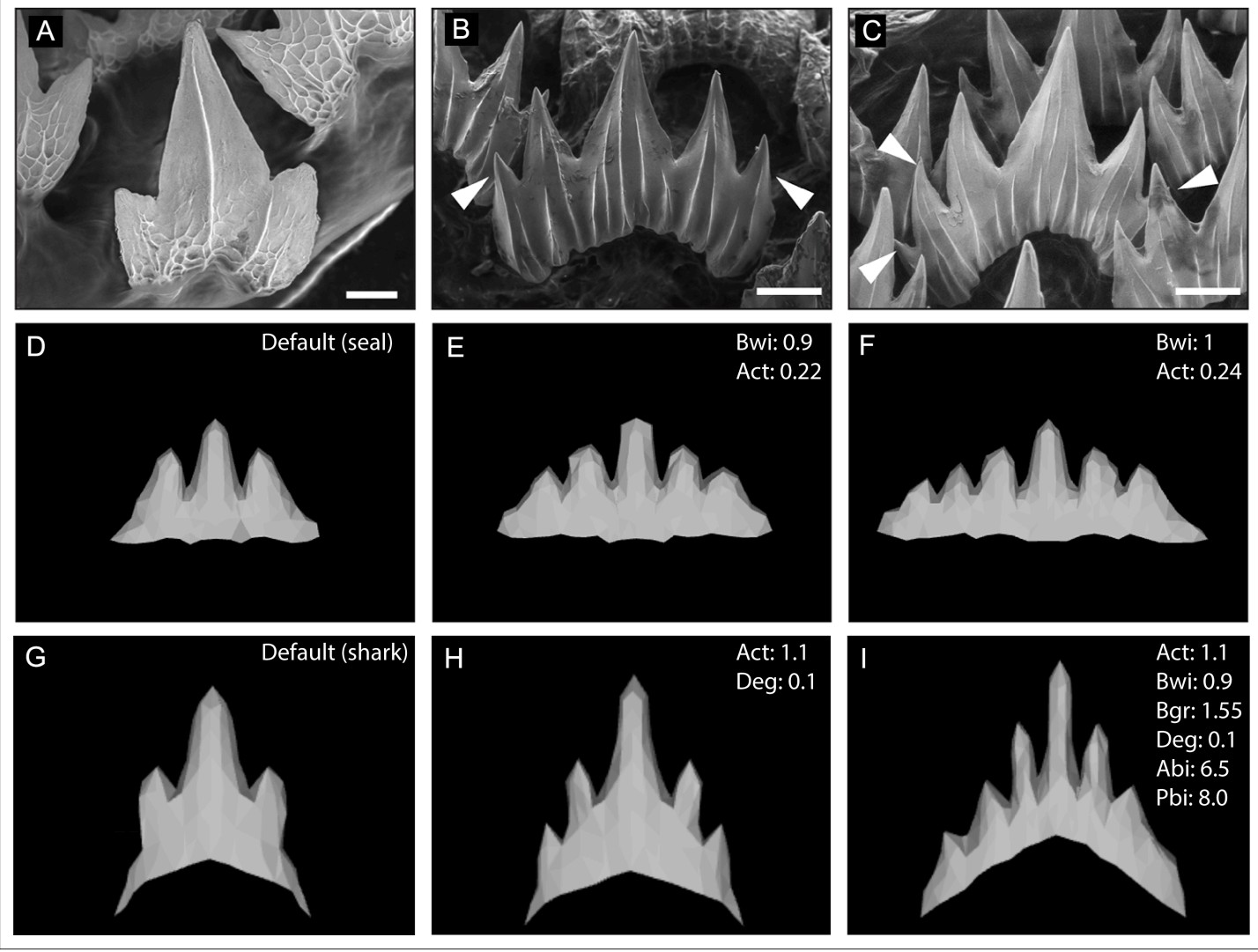

**Figure 8.** In silico modelling of the catshark dentition. Wild-type scanning electron microscope (SEM) images of embryonic (**A**) and juvenile (**B and C**) catshark samples show a shift in cusp number from 3 to 6 cusps during ontogeny. The computational model 'ToothMaker' (*Salazar-Ciudad and Jernvall, 2010*) was used to generate in silico models of the dentition using baseline seal parameters (**D–F**) and parameters refined to produce the characteristic shark dental morphology (**G–I**). Modification to the default seal (D; *Supplementary file 1*: seal) and default shark (G; *Supplementary file 1*: shark) parameters to generate 5-cusped (**E and H**) or 6-cusped (**H–I**) teeth are displayed in the individual panels. 11,000 iterations of the model were run when modelling dentition. White arrowheads represent the addition of extra cusps during ontogeny. Scale bars are 50 µm, and 200 µm in B and C.

inhibitory zone between units, which could represent a constraint on border size. Thus, the shift from 5 to 6 cusps is more complex than increasing the overall level of activation, simply doing so results in additional EKs forming outside of the anterior-posterior axis. To obtain 6-cusp teeth, the anterior-posterior bias can be skewed. Combined with the wider initial border, in this case one larger cusp still sits centrally, as seen in wild-type shark teeth (*Figure 8A–C*).

## In silico modelling of tooth shape following perturbation of Wnt signalling

Following shifts in dental morphology because of canonical Wnt manipulation, we sought to determine which parameters could generate comparable shifts in silico (*Figure 9*). We identified two genetic parameters, which were individually capable of generating comparable phenotypes to the chemically treated samples. Both decreasing the activator auto-activation (Act) (seal baseline: Act = 0.1; shark baseline: Act = 0.25) or increasing the inhibition of the activator (Inh) (seal baseline: Inh = 8; shark baseline: Inh 52) (*Figure 9A and D*) resulted in unicuspid phenotypes strikingly like IWR-1-endo-treated

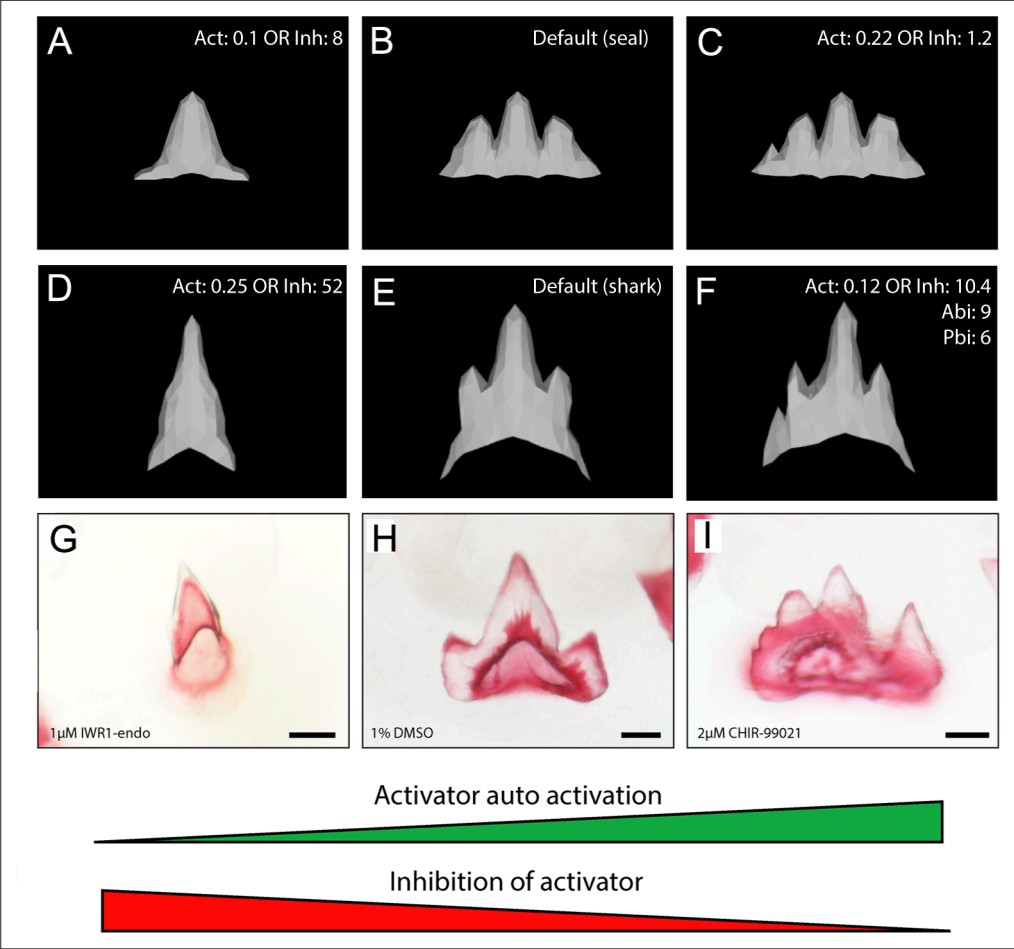

**Figure 9.** In silico modelling recreates catshark tooth phenotypes comparable to canonical Wnt manipulation. In silico modelling using baseline seal parameters (**A–C**) and parameters refined to produce the characteristic shark dental morphology (**D–F**) is also capable of reproducing dental morphologies observed following chemical treatment (**G, I**). 1 μM IWR-1-endo resulted in unicuspid teeth (**G**), whereas 2 μM CHIR9902 (**I**) resulted in the development of supernumerary cusps. DMSO controls exhibit a normal tricuspid dentition (**H**). Either an increase in activator auto-activation (Act) or decrease in inhibition of activator (Inh) is sufficient to shift teeth from a unicuspid (**A,D**) to tricuspid (**B,E**) and quadricuspid (**C**) morphology. Modification to the default seal (B; *Supplementary file 1*: seal) and default shark (E; *Supplementary file 1*: shark) are displayed in the individual panels. 11,000 iterations of the model were run when modelling the effect of small molecule treatment. Scale bars are 50 μm in D–F.

samples (*Figure 9G*). These results are biologically relevant. IWR-1-endo increases the production of the canonical Wnt inhibitor Axin2 (*Chen et al., 2009*). The resulting function of increasing Axin2 in vivo can be equally compared to increasing the inhibitor (Inh) or decreasing the effect of an activator (Act) due to increased inhibition, in silico. The converse is true for CHIR99021, which stabilises cytoplasmic β-catenin leading to upregulated canonical Wnt signalling (*Wagman et al., 2004*). This can be compared to decreasing the effect of an inhibitor, which would lead to higher cytoplasmic β-catenin (seal baseline: Inh = 1.2; shark baseline: 10.4) or increasing the activator itself (seal baseline: Act = 0.22; shark baseline: 0.12) (*Figure 9C*).

The phenotypes generated by the model resulted in both wider teeth and an increase in cusp number. This is directly comparable to the phenotypes observed following treatment with CHIR99021 (*Figure 9I*). In the model, the regulation of the inhibitor and activator stems from the EK signalling centre (*Salazar-Ciudad and Jernvall, 2010*). The recreation of comparable phenotypes through the alteration of biologically relevant model parameters suggests a potential role of canonical Wnt signalling in the regulation of cusp number in the catshark (*Figure 9*). Our findings show that increasing

canonical Wnt activation results in teeth more similar to the adult phenotype, seen through an increase in cusp number following CHIR99021 treatment.

## Discussion

Overall, our results demonstrate that signalling centres comparable to mammalian EKs are present in the shark during tooth morphogenesis. Our data implies that these dental signalling centres function to generate cusps in the shark. Although, how functionally similar these are to mammalian EKs requires further investigation. We identify restricted expression of Fgf markers within the non-proliferative apical dental epithelium corresponding to the same patterns of expression in the mammalian EK. We highlight shifts in cusp number and tooth shape following canonical Wnt manipulation, with similar results simulated via in silico modelling. Of course, care should be taken when interpreting the results of a model compared to real biological signalling, where a model cannot simply account for the complexity of a single pathway. Overall our data imply that a common set of developmental principles account for tooth morphogenesis between mammals and sharks. These common principles include the tip-down development of teeth from an apical signalling centre, and we suggest that Wnt signalling may take part in the activation-inhibition mechanisms that influence cusp patterning related to this signalling centre. Therefore, we suggest that the basic developmental principles of mammalian tooth morphogenesis are shared with sharks, and that perhaps shark tooth shape variation can be modelled with the same developmental principles. Future work will address whether there are indeed shared functional relationships between the mammalian and shark dental signalling centres.

### Sharks possess a comparable EK-like signalling centre to the mammalian EK

Low levels of proliferation at the apical tip of the developing tooth have hinted at the presence of an EK in sharks (*Rasch et al., 2016*), although 3D restriction of signalling molecules within a distinct signalling centre is yet to be described. We note the expression of *fgf10* and the canonical Wnt inhibitor *dkk1* within the non-proliferative dental epithelium (*Figure 4*). Furthermore, our whole mount in situ hybridisation data reveals restricted upregulation of *fgf3* and *fgf10* within both primary (*Figure 2*) and secondary cusp regions (*Figure 5*) and therefore the presence of a spatially restricted dental epithelial signalling centre. In mammals, the expression of EK markers precedes dental shape change (*Vaahtokari et al., 1996*). Similarly, we observe the restricted expression of *fgf3* (*Figure 3F*), *fgf10* (*Figure 4A*), and *shh* (*Figure 2*), very early during dental morphogenesis, prior to the establishment of the overall tooth shape. This suggests that signalling precedes dental shape change and suggests that Fgf and Hh signalling may be driving this process in sharks, as in mammals (*Jernvall et al., 1994*; *Kettunen et al., 2000*).

Even at the earliest initiation of teeth in *S. canicula*, an initial, restricted expression pattern of these three markers (*fgf3*, *fgf10*, and *shh*) demarcates the emergence of the superficial first-generation tooth in the shark (*Figure 2*; *Figure 10*). This superficial and specific unit of epithelial expression appears to resemble initiation of the incisor tooth in the mouse, especially at stages E12.5–13.5 (*Ahtiainen et al., 2016*; *Du et al., 2017*; *Hovorakova et al., 2016*). The similarity between the superficial expression of *fgf3*, *fgf10*, and *shh* in the first-generation tooth in the shark (*Figure 2*) compared to the initiation knot (IK), in the incisor region of the mouse, is striking. In the mouse, the IK is an initial signalling centre similar to the primary EK, that marks a defining position between what will become the epithelial prominences of the vestibular lamina and the dental lamina (i.e., the incisor tooth bud; *Du et al., 2017*; *Hovorakova et al., 2016*). Soon after the expression of *Shh* related to the IK in mice, the primary EK forms separately at the cap stage in the incisor tooth bud (*Du et al., 2017*). A vestibular lamina (the division between the epithelial tissues of the lips and teeth in mammals) does not form in the shark, at least not in *S. canicula*, however, other fish species, for example pufferfish, do have a cleft that separates teeth and fleshy lips (*Thiery et al., 2017*). In the shark, the superficial and initial EK-like structure in the first tooth generation appears similar to the superficial IK in mice. Further investigation of EK markers prior to dental morphogenesis is key to discerning the role and regulation of EK-driven dental morphogenesis in the shark.

Importantly, the mammalian EK itself is unable to respond to the Fgf signals, which it emits as it lacks Fgf receptors (*Kettunen et al., 1998*). Instead, these receptors are found within adjacent dental

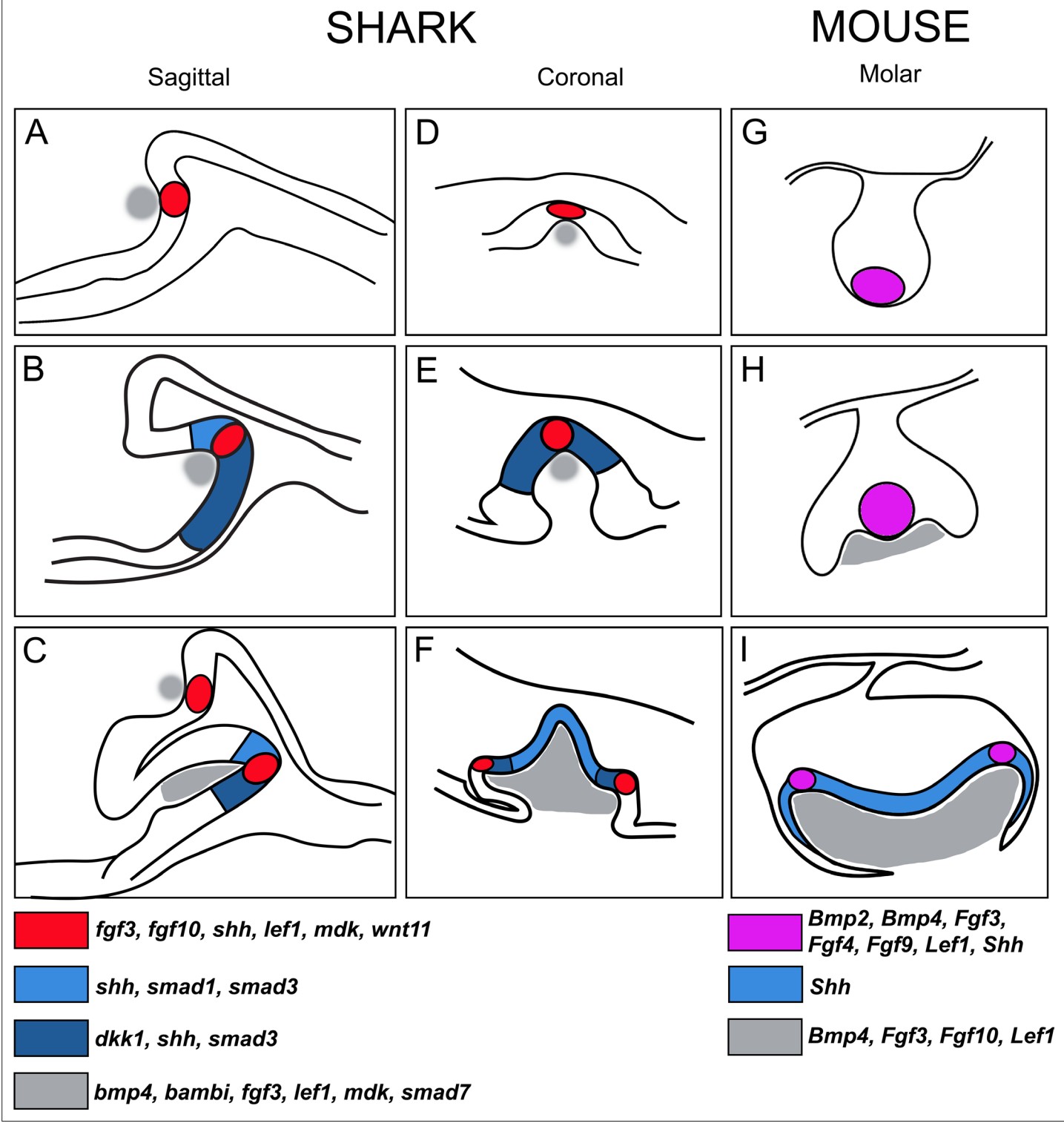

**Figure 10.** Schematic summary of the progression of enamel knot (EK)-like signalling centres in shark teeth versus the EKs during mouse molar development. (**A–C**) Representative sagittal images (based on histological sections) through stages of tooth morphogenesis in the shark (*Scyliorhinus canicula*). (**A**) Initial expression within cells of the pEK-like structure (red; *fgf3*, *fgf10*, *shh*, *lef1*, *mdk*, *wnt11*) in the first, superficial upper jaw tooth position. (**B**) Expression of *shh* expands in the inner dental epithelium beyond the restriction of the EK-like unit (red in A), together with expression of other markers of the IDE (blue), for example, *smad1*, *smad3*, and *dkk1*. Other markers of the apical cells remain restricted, for example, *fgf3*, *fgf10*. (**C**) Further apical signalling occurs as new replacement teeth emerge within the successional lamina. Mesenchymal expression is shown in grey (*bmp4*, *bambi*, *fgf3*, *lef1*, *mdk*, and *smad7*). (**D–F**) Comparative coronal sections through stages of shark tooth morphogenesis, with (**F**) additional secondary

*Figure 10 continued on next page*

*Figure 10 continued*

EK-like signalling centres (sEK) associated with accessory cusps (2 and 3) in the shark tricuspid tooth (red); (**F**) expanded *shh* expression observed in the primary cusp (blue) during later morphogenesis. (**G–I**) Documented expression of genes in the pEK (**G and H**; although these two signalling centres in G and H represent different structures in time and place as described by *Prochazka et al., 2010*; *Mogollón et al., 2021*; *Sadier et al., 2019*) and sEK (**I**) during mammalian molar morphogenesis, showing somewhat equivalent expression and expanded *Shh* expression in the inner dental epithelial cells (IDE; **I**). EK and EK-like units in mouse and shark, respectively, both are restrictive and non-proliferative cell clusters, however, mouse EKs (magenta) are also apoptotic, a character not yet defined in the shark EK-like unit.

epithelial cells, and are important for the induction of differential proliferation between the EK and surrounding epithelial tissue (*Kettunen et al., 2000*). Further research is needed to identify whether this lack of receptors predates the evolution of the mammalian dentition and whether it is fundamentally required for the formation of cusped teeth. Given the conservation of localised Fgf signalling within the shark EK, we hypothesise that a similar lack of receptors drives differential proliferation between the EK and surrounding dental epithelium in the basal gnathostome lineage.

Various developmental characteristics have been used to define the presence of an EK, including spatial restriction of the signalling centre, a lack of cell proliferation, and apoptosis of the epithelial cells within the EK. Apoptosis has been implicated as an important developmental process in regulating the silencing of embryonic signalling centres (*Kettunen et al., 2000*; *Vaahtokari et al., 1996*). Apoptosis has been described in both primary and secondary mammalian EK through TUNEL assays and the expression of the pre-apoptotic marker, p21 (*Jernvall et al., 1998*; *Vaahtokari et al., 1996*). Such assays have so far revealed a lack of apoptosis within the inner dental epithelium of reptiles (*Buchtová et al., 2008*; *Handrigan and Richman, 2010*) and the catshark (*Debiais-Thibaud et al., 2015*) – this has been used to refute the presence of an EK altogether (*Debiais-Thibaud et al., 2015*). However, it has also been shown that mice develop cusps following the loss of apoptosis in caspase-3 deficient mice (*Matalova et al., 2006*). Although morphological defects are identified in non-apoptotic mouse molars (*Matalova et al., 2006*), these results suggest that apoptosis is not fundamentally required in the formation of dental cusps (*Richman and Handrigan, 2011*).

Whilst there are observable differences in signalling between the mammalian and chondrichthyan EKs (*Debiais-Thibaud et al., 2015*), this does not refute the presence of an EK in sharks. Instead, these differences highlight potential lineage-specific modifications which have led to the diversification of vertebrate dental morphology. Given the presence of enameloid, as opposed to 'true' enamel within the Chondrichthyes (*Gillis and Donoghue, 2007*), it has been proposed to term this signalling centre in sharks the 'enameloid knot' (*Rasch et al., 2016*). Our results reveal a complete lack of *bmp4* expression within the non-proliferative apical dental epithelium. *bmp4* expression in surrounding tissues suggests that unlike in mammals, where Bmp4 is secondarily upregulated within the EK, bmp4 may play a role in restricting the expression of other signalling molecules to the EK in the shark. Furthermore, *Shh* is a key marker of the EK in mammals. Although we note its expression within the apical dental epithelium, there is little downregulation of its expression within the inter-cusp dental epithelium. Asymmetric *Shh* expression is thought to regulate polarised growth of the developing feather bud (*Ting-Berreth and Chuong, 1996*) as it does in the zone of polarising activity within the vertebrate limb bud (*Riddle et al., 1993*). It is possible that shh is playing a similar role in teeth of sharks; shh may be regulating polarised growth of the tooth along the oral/aboral axis, but not the medial/lateral axis along which the cusps form. Despite differences in the gene-specific expression patterns of the mammalian and chondrichthyan EKs, representative markers of canonical Wnt, Fgf, Bmp, and Hh signalling are all found expressed within the apical dental epithelium.

## Canonical Wnt signalling as a regulator of natural shark dental variation

Chondrichthyan dental morphology is highly variable between species (*Corn et al., 2016*; *Jambura et al., 2020*). This variability allows for fossil samples to be commonly identified on their dentition alone (*Whitenack † and Gottfried, 2010*). Variation in chondrichthyan tooth shape is typically observable as a modification of the cusp. Examples include: the multi-cuspid saw-like teeth of the sixgill shark (*Hexanchus griseus*); the elongated primary cusp of mako shark teeth (*Isurus oxyrinchus*); serrated teeth of the tiger shark (*Galeocerdo cuvier*); and the ontogenetic shift from tricuspid-heptacuspid teeth of the small-spotted catshark (*S. canicula*) (*Corn et al., 2016*; *Jambura et al., 2020*). Our small molecule manipulations of the canonical Wnt signalling pathway result in a shift in cusp number and

an increase in the variability of tooth shape. Furthermore, the diversity of phenotypes observed following treatment include: unicuspid, multi-cuspid, and serrated-like teeth. Interestingly, the factors and developmental mechanisms that can lead to the formation of serrated teeth in any vertebrate are largely unknown. Restriction of tooth positions in more posterior regions of the jaw could reflect the size shifts in the jaw along a gradient, from the medial to lateral (posterior) sites, which may be related to constraints on cusp number in other vertebrates (*Renvoisé et al., 2017*; *Sadier et al., 2019*; *Savriama et al., 2018*). However, in the catshark at least, the site restriction of potential activatory and inhibitory signals in the oral epithelium may not affect the number of cusps, as this seems less affected. The posterior teeth in the catshark appear to become smaller but retain multiple cusps (*Figure 1*; *Figure 5*).

Canonical Wnt pathway signalling is known to lie upstream of EK signalling during mammalian molar morphogenesis, with its upregulation and downregulation leading to supernumerary EKs and the formation of blunted cusps, respectively (*Järvinen et al., 2006*; *Liu et al., 2008*). In the shark, our findings indicate a similar role for Wnt signalling during dental morphogenesis. Here, Wnt manipulation dramatically disrupts cusp development. As a result, we speculate that alterations to the canonical Wnt pathway may also underlie the natural diversity of tooth shape found between chondrichthyan species. However, although our chemical manipulation phenotypes match what would be expected as a result of disrupted EK signalling, we cannot exclude the possibility that the observed shifts in shape arise as a result of changes in size of the teeth or developmental arrest of tooth morphogenesis. Analysis of canonical Wnt signalling targets within the EK following chemical manipulation could provide further clarity as to which of these developmental processes is driving shape change.

An in silico model of dental morphogenesis developed by *Salazar-Ciudad and Jernvall, 2010*, demonstrates how dental shape can be regulated by a variety of cellular and genetic parameters in the EK. Subtle changes to these parameters lead to a drastic change in dental shape and cusp number. We show that small changes to two genetic parameters regulating activation (*Act*) or inhibition (*Inh*) of the activator in the EK are sufficient in generating teeth representative of upregulation and downregulation of canonical Wnt signalling, respectively. However, the parameters necessary to produce an ontogenetic shift in shark-specific tooth shape may be more complex (*Figure 9*). The similarities observed following Wnt treatments and in silico EK signalling manipulations provide both an element of validation for the in silico model and further evidence of a role for canonical Wnt signalling in the chondrichthyan EK.

## Was an EK-like signalling centre present prior to the evolution of teeth?

Odontodes (tooth-like structures) are thought to have initially evolved outside of the oral cavity, with teeth arising through co-option of the underlying odontode gene regulatory network (*Donoghue and Rücklin, 2016*; *Fraser et al., 2010*; *Martin et al., 2016*). Dermal denticles (non-regenerative odontodes present on the skin surface of chondrichthyans) and teeth are deeply homologous, sharing a high degree of structural and developmental conservation (*Cooper et al., 2018*; *Cooper et al., 2017*; *Martin et al., 2016*). We observe identical expression patterns for *fgf3*, *bmp4*, and *shh*, within and around the apical tip of developing teeth (*Figure 3*; *Figure 5*) and dermal denticles in the catshark (*Cooper et al., 2018*; *Cooper et al., 2017*). As a result, we believe it is likely that the origin of an apical epithelial signalling centre regulating differential epithelial proliferation in epithelial appendages evolved early within the vertebrate lineage and likely predates the evolution of teeth. Therefore, to account for this regulation of shape among disparate developmental structures, we propose a new term to reflect the conservation of this signalling unit, the 'apical epithelial knot' (AEK). This then suggests that this signalling centre is not limited to teeth but can be present in a number of protrusible developmental elements, including non-oral odontodes, for example, skin denticles in sharks (*Figure 1A*). We acknowledge that the term EK should perhaps be reserved exclusively for mammalian tooth signalling. This would allow a distinction to be drawn between mammals and other vertebrates, for example, fishes due to the distinction between the enamel and the enameloid capping mineral layer observed in fish teeth and denticles, and other odontodes. This, however, should not distract from the incredible level of conservation and the shared characters that unite this signalling centre among the vast evolutionary history of vertebrates.

## Conclusion

Although there are differences in the expression of key mammalian EK markers in the shark, there is also a high level of conservation in the expression of key Wnt, Fgf, Bmp, and Hh markers. This level of conservation unites all toothed vertebrates in several ways: (i) the co-expression of a set of common genes; (ii) the patterning and morphogenesis of cusps (i.e., observed between simulations of shark and seal/mammalian teeth); and (iii) the presence of a comparable and functionally similar signalling centre, known in mammals as the EK and now could be collectively be known among vertebrates as an AEK. Contrary to prior assertions, these results provide evidence of an EK signalling centre in an early vertebrate lineage. We therefore suggest that mammalian-specific gene expression patterns within the EK are merely lineage-specific modifications of an ancestral signalling centre (AEK) that likely evolved prior to the appearance of oral teeth.

# Materials and methods

### Animal husbandry

The University of Sheffield is a licensed establishment under the Animals (Scientific Procedures) Act 1986. All animals were culled by approved methods cited under Schedule 1 to the Act. Small-spotted catshark embryos (*S. canicula*) were obtained from North Wales Biologicals, Bangor, UK. Embryos were raised in recirculating artificial seawater (Instant Ocean) at 16°C. At the required stage, embryos were anaesthetised using 300 mg/l MS-222 and fixed overnight in 4% paraformaldehyde at 4°C. Samples were then dehydrated through a graded series of DEPC-PBS/EtOH and kept at –20°C.

### Sectioning and histology

Following dehydration, samples were cleared with xylene and embedded in paraffin. Fourteen μm sagittal sections were obtained using a Leica RM2145 microtome. For histological study, sections were stained with 50% Haematoxylin Gill no.3 and Eosin Y. Slides were mounted with Fluoromount (Sigma) and imaged using a BX51 Olympus compound microscope.

### Scanning electron microscopy

SEM images were obtained using a Hitachi TM3030Plus Benchtop SEM at 15,000 V.

### In situ hybridisation probe synthesis

Protein coding sequences for *S. canicula* were obtained from a de novo transcriptome assembly (Thiery et al., unpublished). Sequences were compared with a range of other vertebrate sequences taken from ensembl.org in order to verify sequence identity. *S. canicula* total RNA was extracted using phenol/chloroform phase separation and cleaned through EtOH/LiCL precipitation. RT-cDNA was made using the RETRO script 1710 kit (Ambion). Probes were made using forward and reverse primers designed through Primer3. Primer sequences are available in *Supplementary file 2*. Probes were chosen to be ~400–800 bp in length. Sequences of interest were amplified from the cDNA through PCR and ligated into the pGEM-T-Easy vector (Promega). Ligation products were cloned into JM109 cells. Plasmid DNA was then extracted from chosen colonies using a Qiaprep spin Mini-prep kit (Qiagen) and sequenced (Applied Biosystems' 3730 DNA Analyser) through the Core Genomics Facility, University of Sheffield. Verified vectors were then amplified through PCR and used as a template for probe synthesis. Sense and anti-sense probes were made using a Riboprobe Systems kit (Promega) and SP6/T7 polymerases (Promega). Probes were labelled with Digoxigenin-11-UTP (Roche) for detection during in situ hybridisation. A final EtOH precipitation step was carried out to purify the RNA probe.

### Section in situ hybridisation

Sagittal paraffin sections were obtained as previously described. Slides were deparaffinised using xylene and rehydrated through a graded series of EtOH/PBS. Slides were incubated in pre-heated pre-hybridisation solution pH 6 (250 ml deionised-formamide, 125 ml 20× saline sodium citrate [SSC], 5 ml 1 M sodium citrate, 500 μl Tween-20, and 119.7 ml DEPC-treated ddH$_2$O) at 61°C for 2 hr. Slides were transferred to pre-heated pre-hybridisation solution containing DIG labelled RNA probe (1:500) and incubated overnight at 61°C. The following day, slides underwent a series of 61°C SSC

stringency washes to remove unspecific probe binding (2 × 30 m 50:50 pre-hybridisation solution: 2× SSC; 2 × 30 m 2× SSC; 2 × 30 m 0.2× SSC). Following the stringency washes, samples were incubated in blocking solution (2% Roche Blocking Reagent [Roche]) for 2 hr at room temperature and then incubated in blocking solution containing anti-Digoxigenin-AP antibody (1:2000; Roche) overnight at 4°C. Excess antibody was washed off through 6 × 1 hr MAB-T (0.1% Tween-20) washes. Slides were then washed in NTMT and colour reacted with BM-purple (Roche) at room temperature and left until sufficient colouration had taken place. Following the colour reaction, a DAPI nuclear counterstain (1 μg/ml) was carried out before mounting the slides using Fluoromount (Sigma). Images were taken using a BX51 Olympus compound microscope. Images were contrast enhanced and merged in Adobe Photoshop.

## Double in situ hybridisation/immunohistochemistry

For double in situ hybridisation/immunohistochemistry, samples first underwent in situ hybridisation as previously described. Immediately after colour reaction, samples were fixed for 1 min in 4% para-formaldehyde in PBS. Samples were then blocked with 5% goat serum and 1% bovine serum albumin in PBS-T (0.05% Tween-20). Blocking solution was replaced with blocking solution containing mouse anti-PCNA primary antibody (ab29; Abcam) at a concentration of 1:2000. Goat anti-rabbit Alexa-Fluor 647 (1:250) (A-20721245; Thermo) and goat anti-mouse Alexa-Fluor 488 (1:250) (A-11-001; Thermo) secondary antibodies were used for immunodetection. Samples were counterstained with DAPI (1 μg/ml) and mounted using Fluoromount (Sigma). Images were taken using a BX51 Olympus compound microscope. Images were contrast enhanced and merged in Adobe Photoshop.

## Whole mount in situ hybridisation

Whole mount in situ hybridisation was carried out in accordance with the section in situ hybridisation protocol with some minor modifications. Following rehydration, samples were treated with 0.2 μg/ml proteinase K for 1 hr at room temperature and then fixed for 20 m in 4% paraformaldehyde in PBS. Samples were then placed in pre-hybridisation and probe solution as previously described. Stringency washes were carried out at 61°C (3 × 30 m 2× SSC-T [0.05% Tween-20]; 3 × 30 m 0.2× SSC-T [0.05% Tween-20]). Blocking, antibody incubation, and colour reaction were carried out as previously described. Following colour reaction, samples were stored in PBS with 10% EtOH.

## Small molecule Wnt perturbations

Ten mM IWR-1-endo (TOCRIS; product no: 3532) and 5 mM CHIR99021 (TOCRIS; product no: 4423) stock solutions were made using dimethyl sulfoxide (DMSO) as a solvent. At ~100 dpf (mid-stage 32), catshark samples were extracted from their egg cases and incubated in 70 ml polypropylene containers. Samples were treated with 1 μM (1:10,000 stock dilution) IWR-1-endo (N=8), 2 μM (1:2,500 stock dilution) CHIR99021 (N=8). 0.1% DMSO was used as a control (N=7). Chemical stock solutions were diluted in artificial seawater with 1% penicillin/streptomycin. Samples were treated with 20 ml of solution, which was replaced every 2 days for a total treatment period of 2 weeks. Following treatment, samples were raised in artificial seawater for a further 4 weeks. After recovery, samples were sacrificed using 300 mg/L MS-222 and fixed overnight in 4% paraformaldehyde at 4°C. Samples were then stained with 0.02% alizarin red in 0.1% KOH overnight in the dark and subsequently cleared in 0.1% KOH. Once residual alizarin red had been removed, samples were transferred into glycerol through a glycerol/0.1%KOH graded series and imaged using a Nikon SMZ1500. Images of chemical perturbation experiments on whole mount jaws are presented in *Figure 6—figure supplements 2–4*.

## Geometric morphometric analysis

Images taken from the small molecule treated clear and stained specimens were analysed using 2D geometric morphometrics. Images of the three most lateral teeth on both the left and right side of the lower jaw were included in the analysis. TPS files containing the treatment images were generated in tpsUtil (*Rohlf, 2009a*; *Rohlf, 2009b*). Landmark coordinates were assigned using tpsDig2 (*Rohlf, 2009a*). Two fixed landmarks were placed on each tooth, one at the tip of the primary cusp and one at the base of the tooth. Eighteen sliding semi-landmarks were then distributed evenly between these two points on either side of the tooth. TPS files were analysed using the R package geomorph (*Adams, 2018*). A GPA was carried out in order to rotate, centre, and rescale image coordinates. Following

GPA, we measured the effect of treatment and sample on centroid size using a linear model, with both factors included as fixed effects. Comparisons between treatments were made using a post hoc Tukey test. Next, we assessed the contribution of treatment and sample to shape (Procrustes aligned coordinates) using a linear model procD.lm function in the geomorph R package (*Adams, 2018*) with both factors included as fixed effects. In order to plot the shape data, a PCA was conducted on the Procrustes aligned co-ordinates and plotted based on the principal components which explain the most variation in the data (*Figure 6C*). We measured the variance for PC1 within treatments and carried out a pairwise F-test to test for changes in variance as a result of treatment.

### ToothMaker modelling of catshark dentition

In silico models of the catshark dentition through ontogeny were generated using the computational model ToothMaker (*Salazar-Ciudad and Jernvall, 2010*), together with the equivalent cusp number changes in the baseline seal tooth parameters (*Salazar-Ciudad and Jernvall, 2010*). Initial shark phenotype baseline parameters were established with assistance from the program creator (Jukka Jernvall, Pers. Comm.). We iterated through each parameter at 10% intervals; using a process of elimination to refine the final wild-type tooth parameters for the modified seal tooth phenotypes to show cusp number equivalence to the catshark (*Supplementary file 1*). Trial and error was then used to determine which parameters were capable of generating teeth resembling those produced following small molecule treatment (*Figure 9*) and in normal development (*Figure 8*). 11,000 iterations of the model were run when modelling the ontogenetic shifts in the dentition.

## Acknowledgements

We are grateful to Jukka Jernvall for his invaluable insights and advice with regard to the ToothMaker simulation software. We thank Keith Hunter and Abigail Tucker for useful comments on previous versions of this manuscript. Funding: University of Sheffield Adapting to the Challenges of Changing Environment Doctoral Training Programme (ACCE) (to RLC and APT); Natural Environment Research Council grant NE/K014595/1 (to GJF); Leverhulme Trust grant RPG-211 (to GJF).

## Additional information

### Funding

| Funder | Grant reference number | Author |
| --- | --- | --- |
| Natural Environment Research Council | NE/K014595/1 | Gareth J Fraser |
| Leverhulme Trust | RPG-211 | Gareth J Fraser |
| University of Sheffield | | Rory L Cooper Alexandre P Thiery |

The funders had no role in study design, data collection and interpretation, or the decision to submit the work for publication.

### Author contributions

Alexandre P Thiery, Conceptualization, Investigation, Writing - original draft, Writing – review and editing; Ariane SI Standing, Rory L Cooper, Investigation, Methodology, Writing – review and editing; Gareth J Fraser, Conceptualization, Funding acquisition, Investigation, Project administration, Supervision, Visualization, Writing – review and editing

### Author ORCIDs

Rory L Cooper http://orcid.org/0000-0003-0172-4708
Gareth J Fraser http://orcid.org/0000-0002-7376-0962

### Decision letter and Author response

Decision letter https://doi.org/10.7554/eLife.73173.sa1
Author response https://doi.org/10.7554/eLife.73173.sa2

## Additional files

### Supplementary files
• Supplementary file 1. ToothMaker parameters. Default ToothMaker parameters set for baseline seal tricuspid tooth and baseline shark tricuspid tooth in *Figures 8 and 9* (*Savriama et al., 2018*). Shifted parameters in red.

• Supplementary file 2. Primer sequences. Primer sequences used to generate the RNA probes for in situ hybridisation in *Figures 2–5* (and *Figure 2—figure supplement 1*).

• Transparent reporting form

### Data availability
All data generated or analysed during this study are included in the manuscript and supporting files.

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
