## [Editor Report]

The manuscript by Thiery et al brings valuable results for understanding the developmental strategies of signalling centres during shark teeth appearance. We believe that it will bring new insight into the general understanding of the phylogenesis of organizers or signalling centres.

---

## [Decision Letter]

**Decision letter after peer review:**

Thank you for submitting your article "An ancient dental signalling centre supports homology of an enamel knot in sharks" for consideration by *eLife*. Your article has been reviewed by 3 peer reviewers, one of whom is a member of our Board of Reviewing Editors, and the evaluation has been overseen by a Reviewing Editor and Didier Stainier as the Senior Editor. The following individual involved in review of your submission has agreed to reveal their identity: Jukka Jernvall (Reviewer #3).

Essential revisions:

1. According to the Reviewer 2 the data support the tip–down development of shark teeth with some critical involvement of the Wnt pathway more than the development of shark teeth from an apical signaling center. Thus, an additional proof would be helpful to show if the gene expressions observed reflect strictly an existence of a signalling center as enamel knot in the mammalian word meaning, it means if both structures are really homological. The reviewers agree that it is expected that all epithelial organs are generally sensitive to modification of Wnt signaling. It is clear in this case, that obtaining new data is limited due to alternative model used in the study. However, please, consider a possible additional experimental focus on Shh pathway, what could be easily done since several inhibitors and activators are available.

2. The main part needing substantial revision is related to Shh expression – Shh expressions in the shark tooth germs seems to form stage specific expression patterns similarly to mice and related to this the authors should discuss the enamel knot like structure in the shark and its relevance to the early placode expressing Shh with its initiating function in the mouse incisor and/or to the later appearing enamel knot with its morphogenetic function as well as to the expression of Shh in the inter–cusp areas in the shark and its possible similarity with Shh expression in the inner dental epithelium at later stages of the mouse odontogenesis.

3. A summarizing scheme would improve the clarity of the gene expressions assessed in the manuscript. A comparison with data available on the mouse would be also helpful.

4. In relation to the point 1, it would also help to revise the manuscript using "signaling centers comparable to mammalian enamel knots in the shark and working similarly as mammalian enamel knots" and avoiding the still not definitely proven existence of „the enamel knot as a morphological structure in the shark".

5. Please, revise the PCA plots and related parts with regard to the comments of reviewers 2 and 3. It would be also interesting to point on the PCA the teeth that are shown later on pictures.

*Reviewer #1 (Recommendations for the authors):*

The authors claimed a difference between early and new generations of teeth in the catshark during development. The early generation develops more superficially in contrast to deeper ingrowing germs of additional generation of teeth. Interestingly, in the mouse incisor it has been documented that the first and initial tooth placode at early stages of the development with its own signalling centre is also formed more superficially in contrast to the subsequently appearing pEK located deeper in the epithelium of the incisor germ. The role of the first placode has been related to the initiation of the functional tooth germ formation (Ahtiainen et al., 2016). pEK is appearing as a structure as late as at E13 at the tip of the deeper ingrowing epithelial bud and consequently the cervical loops start to form around pEK. From this aspect it would be interesting to focus more on the differences between the first generation of teeth present superficially in catshark and the successional generations ingrowing deeper into the surrounding mesenchyme. The first more superficial area of signalling centre formation in the mouse has been homologized with ancestral structures contributing also to the vestibular area (Hovorakova et al., 2016). Would it be possible to compare the expressions of especially epithelial markers shown to play a role in the first superficial teeth with later appearing teeth in the catshark and to discuss this comparison with respect to the knowledge of these two signalling centres with different roles in the mouse?

The authors used the computational in silico model, where the regulation of the inhibitor and activator of the tooth stems from its EK signalling centre. The authors claim in the manuscript that the tooth size and number of cusps is increasing in the new generations of dentition in the catshark, what they relate to the increasing initial site of the tooth formation and what also corresponded with the stimulated model situation. This could be probably connected to the growth of the jaws which gives more space between single signalling centres. Following this space expansion, the changes of the activation and inhibition (of activators and inhibitors levels defunding between neighbouring teeth in the tooth–row as well as between the tooth generations) could be the cause of this change in size and shape. The lower inhibition of neighbouring teeth could, as a consequence, allow the signalling centres to activate themselves more and expand more in size and produce bigger teeth with additional cusps. As shown in the mouse molar tooth row, the space limits are one of the main factors of activation–inhibition working during antero–posterior signalling centres appearance (Sadier et al., 2019). The authors are mentioning the gradient of activators and inhibitors in the culture. Maybe this model situation reflects more the reality of the activator–inhibitor gradients in the jaw? This should be discussed in the Discussion.

The authors documented that Shh, which is a key marker of EK in mammals including mice, is expressed within the apical dental epithelium in the catshark. However, Shh expression was downregulated between cusps within the inter–cusp dental epithelium. In the mouse Shh is not limited only to EK. Its expression is stage dependent and in more advanced stages its expression is expanding also to inter–cusp dental epithelium, to the whole inner dental epithelium, resp. inner enamel epithelium, in fact. Is this a real downregulation in the inter–cusp area of the dental epithelium in the catshark? Could not be this situation similar to the mouse IEE resp. IDE? According to the Figures E, F the Shh expression seems to be region specific and it changes antero–posteriorly. It could reflect the antero–posterior appearance of the tooth germs. The posterior ones would be less advanced and thus Shh expression in the anterior part of the tooth–row would be upregulated in fact. This should be discussed with respect to the suggested role of Shh regulating polarised growth of the tooth along the oral/aboral axis.

*Reviewer #2 (Recommendations for the authors):*

– The expression pattern of many genes was assessed. A summarizing scheme including the mammalian expression when available would have been much helpful to make the best of these data.

– The effect of treatments was quantified with 2D–outline analysis and statistics. Because they seem to produce opposite effects, samples with opposite treatments were grouped together. It would however be appropriate to also treat samples separately according to the treatment, even though significativity might not be reached because of limited sample size. It would be nice to point on the PCA the teeth that are shown later on pictures. Showing a superposition of outlines could also help the reader to get a general appreciation of treatments' effects.

– This bias may result from accessibility of the drug through the tissue: This sounds weird to me. Why should it be a property of the treatment and not simply that the treatment reveals a property of the tissue?

– "Similarly, we observe the restricted expression of *fgf3* (Figure 2F) and fgf10 (Figure 3A) very early during dental morphogenesis, prior to the establishment of the overall tooth shape". OK for *Fgf3*, but for FGF10, I could not find a picture where primary cusp morphogenesis is already ongoing??

– "EK signaling" seems to be used for "EK patterning" or "EK formation" at several places. Ex: "Wnt signalling appears to be a primary upstream regulator of EK signaling"

*Reviewer #3 (Recommendations for the authors):*

1) I sympathize with the authors focus on Wnt signaling in the study, but somehow the argument why this focus was chosen is a bit mumbled. This raises the question why only Wnt signaling has been modified, and not, for example Shh which is known to produce comparable effects in the mouse (in fact, the unifying feature for the genes required for the progression of tooth development may be their high expression level, DOI: 10.1002/jez.b.23009). Moreover, it is pretty much to be expected that all epithelial organs are sensitive to modification of Wnt signaling. What I would suggest to slightly expand the introduction (Page 2, third paragraph starting "Whilst over a dozen markers have been described…"). A central point of canonical Wnt signaling in teeth is that it has different roles in epithelium and mesenchyme. A good reference outlining in its introduction why Wnts can be considered upstream in tooth development is https://doi.org/10.1242/dev.158048.

2) There are places containing inaccuracies about gene expression in mice. Most notably Shh is not only expressed in the enamel knots as currently stated (Page 6, "…shh is confined to the EK in mammalian molars…"). As the development proceeds, Shh expression spreads throughout the inner enamel epithelium (hence the wide use of ShhGFP mice to study tooth development). The results of Thiery et al. on Shh in shark teeth actually fit quite well with the dynamic spreading of Shh in the mouse. For example, see https://pubmed.ncbi.nlm.nih.gov/12403705/ or doi:10.1038/nature06153

The discussion about Bmp4 expression is also a bit confusing. In the mouse, Bmp4 is only expressed in the enamel knot when those cells are about to enter apoptosis. It is true it is expressed in the mesenchyme adjacent to the EK and is required for the induction of the EK. As the shark teeth are much pointier than the mouse molars, it could still be that Bmp4 is involved in the induction of later forming cusps. Currently the interpretation is the opposite, and I leave it to the authors to decide how to address this (Page 4, "The precise exclusion of bmp4 from the non–proliferative tooth apex (Figure 3D and E) raises the possibility that instead of regulating putative EK signalling, it is involved in the restriction of other EK markers.").

Not sure it is correct to call Msx a pathway. It is a transcription factor (homebox gene family) that integrates multiple pathways (Page 6 " Wnt/β–catenin signalling has been implicated upstream of key pathways (Bmp, FGF and Msx) regulating dental morphogenesis [12,13]")

3) Wnt signaling manipulations: There are three places where the phenotypic results are stated to be 'dramatic' (pages 7, 12). Typically, when one needs geometric morphometrics to illustrate the results, the patterns are not very dramatic. In fact, looking at the PCA plot, the results are relatively subtle. I am also not following what is the difference between the main and supplement figure PCA (the main figure seems to be using a mean of the teeth)? Perhaps density ellipses for different treatments would help to illustrate the effects better, as they are there. And a more communicative adjective that could be used to describe the results is, for example, 'substantial'.

4) The reference for the computational model using the ToothMaker is http://dx.doi.org/10.1098/rsos.180903

This paper contains the used model version and the link to the github to obtain the package, and also provides the parameters for the basic tricusped seal teeth (Page 9, "After modelling a wild–type tricuspid tooth in correspondence with the software developers (J. Jernvall, pers. comm.)…"

5) Finally, I wonder if a schematic illustration comparing mammalian and shark EKs would help to communicate the results, and also to point to issues that remain to be explored?

[Editors' note: further revisions were suggested prior to acceptance, as described below.]

Thank you for resubmitting your work entitled "An ancient dental signalling centre in sharks supports homology of tooth shape among vertebrates" for further consideration by *eLife*. Your revised article has been evaluated by Didier Stainier (Senior Editor) and a Reviewing Editor.

The manuscript has been improved but there are some remaining issues that need to be addressed, as outlined below:

1. In their reply to point 1, the authors focused strictly their reply on the suggestion to add experiments on the Shh pathway, but according to Reviewer 2 they did not reply to the first part. A few but mandatory text changes are necessary. This is detailed in the Reviewer 2 recommendations.

2. There is a lack of support of the data showing that dental signaling centers are functionally similar, and the shark EK regulates the formation of cusps. The writing should be revised to tone down this and leave such conclusion for further functional studies. This should be revised and reworded according to reviewer 2 since the authors stated in their response and added this statement in the discussion (page 14, line 672) that "in the catshark at least, the site restriction of potential activatory and inhibitory signals of the oral epithelium may not affect the number of cusps….." Thus, the functional similarity in regulating the formation of cusps is not proven in the catshark and is not supported by the data.

3. The title should be changed as suggested by reviewer 2: "An epithelial signaling centre in sharks supports homology of tooth morphogenesis in vertebrates."

4. The schematics 10 added by the authors should be revised according to the suggestion of reviewers 1 and 2, showing that the signalling centres in G and H reflect different structures in time and in position (for details see Prochazka et al., PNAS, 2010).

*Reviewer #1 (Recommendations for the authors):*

Thank you for the revised version of the paper "An ancient dental signalling centre in sharks supports homology of tooth shape among vertebrates" submitted by Thiery et al.

The authors revised their manuscript substantially according to the suggested points and in my point of view the revisions increased the quality of the manuscript substantially.

I appreciate the change of the title of the manuscript, which now reflects much more the main focus of the paper and avoids confusion in homology of enamel knot in the mammals and enamel knot like structure in sharks.

I also appreciate adding of the schematics as Figure 10, even if we would like to be precise, the situation of the mouse molar germ would probably from a sagittal view show a positional discrepancy between the signalling centres in G and H, since those two centres are not located in the same position of the epithelium, they differ in time and position as shown already by Prochazka et al., PNAS, 2010.

Basically, all the addressed points have been sufficiently responded by authors and I do not have any additional comments.

*Reviewer #2 (Recommendations for the authors):*

Overall, the authors nicely take into consideration a number of points and I feel the manuscript has been improved. I have a major concern about a point that was not taken into consideration (1).

1) About the general conclusion of the paper

Essential revision point 1 was:

"According to Reviewer 2 the data support the tip–down development of shark teeth with some critical involvement of the Wnt pathway more than the development of shark teeth from an apical signaling center. Thus an additional proof would be helpful to show if the gene expressions observed reflect strictly an existence of a signalling center as enamel knot in the mammalian word meaning, it means if both structures are really homological. The reviewers agree that it is expected that all epithelial organs are generally sensitive to modification of Wnt signaling. It is clear in this case, that obtaining new data is limited due to alternative model used in the study. However, please, consider a possible additional experimental focus on Shh pathway, what could be easily done since several inhibitors and activators are available. "

In their reply to point 1, the authors focused strictly their reply on the suggestion to add experiments on the Shh pathway, but they did not reply to the first part. Therefore, I am not satisfied at all by the reply – this is not such a major issue however, since I believe only a few but mandatory text changes are needed to satisfy my concerns. This is detailed below.

As already said in my review, the results obtained by the authors are very valuable, and surely an important step towards a recognition of some homology of tooth shape formation in vertebrates, but they surely do not definitely demonstrate that a signaling center works in sharks as it does in mammals.

This inaccurate claim is based on an unappropriated use of modeling, which needs correction (see below). The experimental data clearly support a role of Wnt signaling in cusp development. The gene expression data are clearly in favor of tip–down development with an apical epithelial signaling center. The mammalian model with minor modifications can recapitulate shark different tooth morphologies and can (more or less) recapitulate morphological changes upon increased/decreased activation of the Wnt pathway. Some aspects are missed however (the recapitulation of Wnt–increased cusp number is far from being perfect). This tels us that shark teeth and their variation can be modeled with the developmental principles of mammalian teeth, and this further suggests that basic developmental principles of tooth morphogenesis are shared with sharks. But this cannot be taken as a definitive proof that an apical signaling center drives cusp formation as in mammals, nor that Wnt signaling takes part to this EK formation. This can surely be discussed in the discussion, but this is surely not a result from the model to be taken for granted. Overall, the data argue for common developmental principles between shark and mammals, including tip–down development with an apical signaling center, and suggest that Wnt signaling, as in mammals, may take part into activation–inhibition mechanisms responsible for the patterning of this signaling center. A plausible interpretation is not definitive proof, and it is important to keep the distinction for future research.

I recommend that several parts of the text are deleted (marked **…** below) or revised to stick the conclusions to the data. The text mentioned below should be removed, and the authors could expand the discussion of their results with more caution.

As I will further detail below, one big issue in the reasoning is that there is no direct correspondence between model parameter and real–life signaling pathways – even in the mammal model. The tooth model is accounting for developmental principles, that's all, and that's already a lot.

– P10. Line 480 implicate canonical Wnt in the regulation of cusp development **and therefore possibly in EK signalling**.

Leave **…** for discussion.

–P11 Line 510–513 **Both properties could be reflective of 510 changes to underlying Wnt signalling, with increased activator/decreased inhibitor directly, or 511 a faster turnover of components favouring activation. Protein degradation is a key component 512 of Wnt signalling activity, of which the GSK3β complex which degrades β–catenin is just one 513 example**

These comments should surely not be not part of the result section. I strongly recommend to totally remove them: the tooth model has not the level of molecular realism that would make this comment meaningful. In mammals, many different pathways but the Wnt pathway have been linked to the activation/inhibition parameters of the model (Shh, Bmp4, Eda…), but no study pretends that one of this pathway "is the parameter". The above attempt to stick to molecular reality is artificial, and moreover, not accurate. Indeed, protein degradation is not taken by the authors as meant in the original model, which is inspired from Turing models = degradation should be the degradation of the activator when diffusing in space, not the degradation of an intracellular component of a signaling pathway.

P11 line 521–524 Thus, the shift from 5 to 6–cusps is more complex 521 than increasing the overall level of **Wnt** activation, simply doing so results in additional 522 enamel knots forming outside of the anterior–posterior axis. **These intricacies over shape 523 control are perhaps enabled by the complexity of Wnt signalling, the number of interacting 524 molecules permitting a wider range of phenotypes.**

Again, these parts should surely not be not part of results, and should be removed in my opinion. There is a misunderstanding on what a model does and does not, and what a model can say or cannot say. The discrepancy between shark perturbed teeth and simulations should – first of all– be taken as an indication that the tooth model does not account for a key principle of shark tooth development – rather than it is missing the molecular complexity of a single pathway. Surely it does miss molecular complexity (that's what we ask to a model) – the question is, does it still capture some general principle (e.g. where additional cusp can form), or not. The authors should acknowledge and discuss more appropriately the discrepancy in the discussion part.

**In the model, the 550 regulation of the inhibitor and activator stems from the EK ignaling centre (Salazar–Ciudad and Jernvall, 2010). The recreation of comparable phenotypes through the alteration of biologically relevant genetic MODEL parameters supports our findings, which demonstrate a critical role of canonical Wnt ignaling in the regulation of cusp number from the catshark EK (Figure 9). **

No, I strongly disagree with this conclusion for the reasons mentioned above. This text should be modified.

Line 560–568

Overall, our results demonstrate that signalling centres comparable to mammalian enamel knots are present in the shark during tooth cusp morphogenesis. **Our data implies that these dental signalling centres are functionally similar in both sharks and the mammals. The enamel knot (EK)–like signalling centre regulates the formation of dental cusps in the catshark with canonical Wnt signalling playing a significant role in this process. **

Again, I disagree that the data show that dental signaling centers are functionally similar, and the shark EK regulates the formation of cusps. The writing should be revised to tone down this and leave such conclusion for further functional studies.

2) About the new title:

"An ancient dental signalling centre in sharks supports homology of tooth shape among vertebrates".

Is it really a matter of tooth shape homology? – I guess the focus is rather the homology of developmental principles? It is unclear to me what for "ancient" stands for: "old like early vertebrate teeth" or "older than teeth themselves". In the first case, it is redundant with the idea of homology in vertebrates, in the second case, I am not sure that the two ideas can be conveyed at the same time in the title. For all those reasons, I would suggest something simple like:

– An epithelial signaling centre in sharks supports homology of tooth morphogenesis in vertebrates

3) About the revised section on gene expression

Please consider improving the writing of the "Key markers of tooth morphogenesis in sharks" section. This long description lacks clear organization for me – I missed the line. For example: *Fgf3*, Fgf10, Shh are introduced lines 191–198, but *fgf3* and fgf10 are discussed again l 237–251, between a Wnt– and a Bmp4– paragraph. At the very end of the section, these 3 genes are mentioned again with this statement, which sounds like a repetition:

" *fgf3*, fgf10, sonic hedgehog (shh)midkine (mdk) have previously been observed within the apical tip of developing teeth in the catshark (Scyliorhinus canicula) and little skate(Leucoraja erinacea), corresponding to the putative EK–like signalling centre (Rasch et al.,299 2020, 2016).

There might be a good reason why mdk is discussed in this very last part, but at least revise the text to put the focus on mdk: “mdk has been previously associated with *fgf3*, fgf10 and shh in”…"Same thing, line 30”: "Another marker, isl1, is expressed (Figure 3P) in a very similar and restricted pattern to fg”10".

It would really help me if all these overlapping gene expression patterns are mentioned together in the same paragraph.

I found figure numbering quite confusing and it seems to me it does not correspond to the text order. Why are Bmp4 results (coming very late in the section) associated to Figure 2 (*fgf3*,fgf10,shh)?

4) About the new carton/figure 10:

The younger cartoon represents a mouse molar bud stage, just before cap transition. At this stage, the first molar EK has just formed and is about to drive cap transition. Bmp4 is expressed in the PEK at this stage (Prochazka et al. 2010).

Therefore, given the color code, the mouse EK should thus be pink, and not red.

*Reviewer #3 (Recommendations for the authors):*

I find the revised manuscript much improved. While one could pick on individual parts, the cumulative evidence provided by the work is compelling. Moreover, the work should stimulate many new studies, both experimental and comparative. I like the 'apical epithelial knot' term proposed.

Below a few details to be fixed.

– For consistency, Sostdc1 should be used instead of Ectodin for the gene name (both mentioned in the text but ectodin used in figures). In fact, Sostdc1 has also names 'wise' and 'Usag1'.

– Supplementary Table S1 should refer to Savriama et al. 2018. Incidentally, I noticed that quite a few parameters have been tinkered with to produce the 'seal' morphology. Although some changes were quite likely unnecessary, I checked and the result is a seal morphology.

– Reference list should be checked. At least Hallikas et al. 2021 is missing.

– Figure 6 (tooth shape analyses) is challenging to read, especially with the very long caption. There are color codes for the different treatments, but especially Figure 6D (PCA plot) is confusing as one tooth is circled with each color. How about trying to use colored 95 or 99% density ellipses for each treatment? Moreover, the Figure 6A has red–green color code that is confusing as these colors do not mean the same as in the rest. Is the 6A needed?

---

## [Author Response]

Essential revisions:1. According to the Reviewer 2 the data support the tip–down development of shark teeth with some critical involvement of the Wnt pathway more than the development of shark teeth from an apical signaling center. Thus, an additional proof would be helpful to show if the gene expressions observed reflect strictly an existence of a signalling center as enamel knot in the mammalian word meaning, it means if both structures are really homological. The reviewers agree that it is expected that all epithelial organs are generally sensitive to modification of Wnt signaling. It is clear in this case, that obtaining new data is limited due to alternative model used in the study. However, please, consider a possible additional experimental focus on Shh pathway, what could be easily done since several inhibitors and activators are available.

We thank the reviewers for this suggestion. And we have carefully considered these additional experiments, however, as is the nature of working with alternative models, we currently have limited access to embryos at the correct stages required to undertake these lengthy experiments, over slow developmental stages. Although we would like to have this data, we cannot perform these experiments easily, and are unfortunately just beyond the scope of this manuscript. We have previously attempted these inhibition experiments and there is an issue with solubility of chemical inhibitors in cold (12-14°C) saltwater. Of course, we continue to attempt new and modified methods to answer questions of functional regulation of signalling pathways. We hope the reviewers can appreciate these experimental hurdles. We have however, included a new figure and further description that we think improves the manuscript, overall.

2. The main part needing substantial revision is related to Shh expression – Shh expressions in the shark tooth germs seems to form stage specific expression patterns similarly to mice and related to this the authors should discuss the enamel knot like structure in the shark and its relevance to the early placode expressing Shh with its initiating function in the mouse incisor and/or to the later appearing enamel knot with its morphogenetic function as well as to the expression of Shh in the inter–cusp areas in the shark and its possible similarity with Shh expression in the inner dental epithelium at later stages of the mouse odontogenesis.

We thank you for the helpful comments here. We have added the following section(s) below to expand the description of shh in the results and discussion, added reference to the murine condition, including the initiation of the incisor, the initiation knot (IK) associated with the division of epithelia between the vestibular and dental lamina, and the striking similarities between the shark and mouse expression patterns. In addition, we have made a new figure (Figure 2) and included an additional schematic figure (Figure 10).

Text added:

Page 4.

“First, we sought to characterise the expression of genes from known pathways (FGF and Hh) associated with initial patterning of tooth cusps in mice (Kettunen et al. 2000; Ahtiainen et al. 2016; Du et al. 2017). Our results show the expression of three genes, *fgf3*, fgf10 and shh, all associated with early restricted epithelial expression in the first, superficial generation of teeth (Figure 2A, D, G), and later in morphogenesis (Figure 2B,C,E,F,H,I; Figure 3G,H) as tooth shape progresses, with the exception of *fgf3* which is also expressed in the underlying mesenchymal apex of the dental papilla (Figure 2A-C). Importantly, the superficial nature of expression in these three genes is associated with the initiation of the first tooth placode (Figure 2A,D,G), and this could in fact mark the onset of the primary EK-like signalling in the shark tooth (see Discussion for further comparative information).”

Page 7.

“As with our section in situ hybridisation data, bmp4 is excluded from the epithelium at the leading edge of the tooth, in whole mount. Instead, its expression appears to be primarily restricted to the dental papilla in both upper (Figure 4A) and lower jaws (Figure 4B). In contrast, shh expression can be seen within the dental epithelium and specifically upregulated within the leading edge of the tooth (Figure 4E and F; Figure 2). shh is confined to the EK in mammalian molars (Hardcastle et al., 1998; Vaahtokari et al., 1996), before spreading throughout the inner enamel epithelium later during tooth morphogenesis (Gritli-Linde et al., 2002; Kavanagh et al., 2007), and interestingly we see strong complementary expression within both the primary cusp (Figure 4E and F: black arrowhead) and later morphogenesis in the shark cusps (Figure 4E and F: white arrowhead; Figure 2). Depending on the stage of morphogenesis, the extent of shh expression within the inter-cusp dental epithelium is variable (See Figure 2). Alongside playing a putative role in EK signalling, the expression of shh within the epithelium at the leading edge of the tooth suggests that it may also be involved in establishing an anterior to posterior growth gradient within the tooth unit. In whole mount, we observe an additional gradient of tooth stages that are reflected in the variation in observed shh expression, with more posterior tooth positions along the jaw arc at earlier stages of morphogenesis (Figure 4E and F). More anterior teeth show apical expression of shh in the secondary cusps, whereas the more posterior tooth positions have shh expression associated with initial tooth buds, primary cusp morphogenesis, and even the inter-tooth epithelium, where new tooth positions are being initiated (Figure 4E” and F”).”

Page 13.

Discussion: “Even at the earliest initiation of teeth in Scyliorhinus canicula, an initial, restricted expression pattern of these three markers demarcates the emergence of the superficial first-generation tooth in the shark (Figure 2; Figure 10). This superficial and specific unit of epithelial expression appears to resemble initiation of the incisor tooth in the mouse, especially at stages E12.5-13.5 (Ahtiainen et al. 2016; Du et al. 2017; Hovorakova et al. 2016). The similarity between the superficial expression of *fgf3*, fgf10 and shh in the first-generation tooth in the shark (Figure 2) compared to the initiation knot (IK), in the incisor region of the mouse, is striking. In the mouse, the IK is an initial signalling centre similar to the primary EK, that marks a defining position between what will become the epithelial prominences of the vestibular lamina and the dental lamina (i.e., the incisor tooth bud; (Du et al., 2017; Hovorakova et al., 2016)). Soon after the expression of Shh related to the IK in mice, the primary EK forms at the cap stage in the incisor tooth bud (Du et al., 2017). A vestibular lamina (the division between the epithelial tissues of the lips and teeth in mammals) does not form in the shark, at least not in S. canicula, however, other fish species e.g., pufferfish do have a cleft that separates teeth and fleshy lips (Thiery et al., 2017). In the shark, the initial EK-like structure in the first tooth generation (similar to the superficial IK in mice) appears to be equivalent to the mammalian primary EK, perhaps not requiring a junction to divide a vestibular and dental lamina permits the development of the superficial first-generation tooth in shark to pattern in place (Figure 2). Further investigation of EK markers prior to dental morphogenesis is key in discerning the role and regulation of EK driven dental morphogenesis in the shark.”

3. A summarizing scheme would improve the clarity of the gene expressions assessed in the manuscript. A comparison with data available on the mouse would be also helpful.

We have now added a new summary schematic figure (Figure 10) – this figure now summarizes the expression data in the shark in both sagittal and coronal view, compared to the known EK expression in the mouse.

4. In relation to the point 1, it would also help to revise the manuscript using "signaling centers comparable to mammalian enamel knots in the shark and working similarly as mammalian enamel knots" and avoiding the still not definitely proven existence of „the enamel knot as a morphological structure in the shark".

Thank you for this comment. We agree. We have taken care to address the nature of conservation. We have changed the title slightly to ease the use of the term homology in direct association with the enamel knot signalling centre to “An ancient dental signalling centre in sharks supports homology of tooth shape among vertebrates” and also edited the first paragraph of the Discussion with the following statement:

Page 12.

“Overall, our results demonstrate that signalling centres comparable to mammalian enamel knots are present in the shark during tooth cusp morphogenesis. Our data implies that these dental signalling centres are similar in both sharks and the mammals. The enamel knot (EK)-like signalling centre regulates the formation of dental cusps in the catshark with canonical Wnt signalling playing a significant role in this process. We identify restricted expression of FGF markers within the non-proliferative apical dental epithelium corresponding to the same patterns of expression in the mammalian EK. We highlight shifts in cusp number and tooth shape following canonical Wnt manipulation, with similar results simulated via in silico modelling.”

Then, the first section of the discussion has an edited subheading:

“Sharks possess a comparable enamel knot-like signalling centre to the mammalian EK”.

And also in the final paragraph we have edited this statement to reflect this point:

“Conclusion

Although there are differences in the expression of key mammalian EK markers in the shark, there is also a high level of conservation in the expression of key Wnt, FGF, Bmp, and Hh markers. This level of conservation unites all toothed vertebrates in several ways: (i) the co-expression of a set of common genes; (ii) the patterning and morphogenesis of cusps (i.e., observed between simulations of shark and seal/mammalian teeth); and (iii) the presence of a comparable and functionally similar signalling centre, known in mammals as the enamel knot (EK) and now could be collectively known among vertebrates as an apical epithelial knot (AEK). Contrary to prior assertions, these results provide evidence of an EK signalling centre in an early vertebrate lineage. We therefore suggest that mammalian-specific gene expression patterns within the EK are merely lineage specific modifications of an ancestral signalling centre (AEK) that likely evolved prior to the appearance of oral teeth.”

5. Please, revise the PCA plots and related parts with regard to the comments of reviewers 2 and 3. It would be also interesting to point on the PCA the teeth that are shown later on pictures.

Thank you for this suggestion. We have now added new annotation to the PCA plot with examples of the relevant tooth shapes (see new Figure 6).

Reviewer #1 (Recommendations for the authors):The authors claimed a difference between early and new generations of teeth in the catshark during development. The early generation develops more superficially in contrast to deeper ingrowing germs of additional generation of teeth. Interestingly, in the mouse incisor it has been documented that the first and initial tooth placode at early stages of the development with its own signalling centre is also formed more superficially in contrast to the subsequently appearing pEK located deeper in the epithelium of the incisor germ. The role of the first placode has been related to the initiation of the functional tooth germ formation (Ahtiainen et al., 2016). pEK is appearing as a structure as late as at E13 at the tip of the deeper ingrowing epithelial bud and consequently the cervical loops start to form around pEK. From this aspect it would be interesting to focus more on the differences between the first generation of teeth present superficially in catshark and the successional generations ingrowing deeper into the surrounding mesenchyme. The first more superficial area of signalling centre formation in the mouse has been homologized with ancestral structures contributing also to the vestibular area (Hovorakova et al., 2016). Would it be possible to compare the expressions of especially epithelial markers shown to play a role in the first superficial teeth with later appearing teeth in the catshark and to discuss this comparison with respect to the knowledge of these two signalling centres with different roles in the mouse?

We have now added a new figure (Figure 2) that shows the transition from (i) the superficial first tooth, to (ii) early and (iii) late morphogenesis. We include the expression of *fgf3*, fgf10 and shh across this transition. We have also included new text to explain this similarity between the incisor signalling centre(s) IK and pEK and the more superficial first-generation teeth in sharks. See comment 2 above.

The authors used the computational in silico model, where the regulation of the inhibitor and activator of the tooth stems from its EK signalling centre. The authors claim in the manuscript that the tooth size and number of cusps is increasing in the new generations of dentition in the catshark, what they relate to the increasing initial site of the tooth formation and what also corresponded with the stimulated model situation. This could be probably connected to the growth of the jaws which gives more space between single signalling centres. Following this space expansion, the changes of the activation and inhibition (of activators and inhibitors levels defunding between neighbouring teeth in the tooth–row as well as between the tooth generations) could be the cause of this change in size and shape. The lower inhibition of neighbouring teeth could, as a consequence, allow the signalling centres to activate themselves more and expand more in size and produce bigger teeth with additional cusps. As shown in the mouse molar tooth row, the space limits are one of the main factors of activation–inhibition working during antero–posterior signalling centres appearance (Sadier et al., 2019). The authors are mentioning the gradient of activators and inhibitors in the culture. Maybe this model situation reflects more the reality of the activator–inhibitor gradients in the jaw? This should be discussed in the Discussion.

We have added a comment about this A-I gradient. We see a reduction in the size of teeth from anterior to posterior positions.

We have now included a statement in the ‘Results’ and ‘Discussion’ to address these comments. The reviewer is quite right in that the more posterior tooth germs are later to form due to antero-posterior wave of tooth initiation, and these are indeed less advanced, and will show differences in the regional specification of shh expression.

Page 8

“In whole mount, we observe an additional gradient of tooth stages that are reflected in the variation in shh expression, with more posterior tooth positions along the jaw arc at earlier stages of morphogenesis (Figure 5E and F). More anterior teeth show apical expression of shh in the secondary cusps, whereas the more posterior tooth positions have shh expression associated with initial tooth buds, primary cusp morphogenesis, and even the inter-tooth epithelium, where new tooth positions are being initiated (Figure 5E” and F”).”

Page 15.

“Restriction of tooth positions in more posterior regions of the jaw could reflect the size shifts in the jaw along a gradient, from the medial to lateral (posterior) sites, this may be related to constraints on cusp number in other vertebrates (Renvoisé et al., 2017; Sadier et al., 2019; Savriama et al., 2018). However, in the catshark at least, the site restriction of potential activatory and inhibitory signals in the oral epithelium may not affect the number of cusps, as this seems less affected. The posterior teeth in the catshark appear to become smaller but retain multiple cusps.”

The authors documented that Shh, which is a key marker of EK in mammals including mice, is expressed within the apical dental epithelium in the catshark. However, Shh expression was downregulated between cusps within the inter–cusp dental epithelium. In the mouse Shh is not limited only to EK. Its expression is stage dependent and in more advanced stages its expression is expanding also to inter–cusp dental epithelium, to the whole inner dental epithelium, resp. inner enamel epithelium, in fact. Is this a real downregulation in the inter–cusp area of the dental epithelium in the catshark? Could not be this situation similar to the mouse IEE resp. IDE? According to the Figures E, F the Shh expression seems to be region specific and it changes antero–posteriorly. It could reflect the antero–posterior appearance of the tooth germs. The posterior ones would be less advanced and thus Shh expression in the anterior part of the tooth–row would be upregulated in fact. This should be discussed with respect to the suggested role of Shh regulating polarised growth of the tooth along the oral/aboral axis.

Thank you for these suggestions, we have now addressed these points. See earlier responses 2 and 6 above.

Reviewer #2 (Recommendations for the authors):The expression pattern of many genes was assessed. A summarizing scheme including the mammalian expression when available would have been much helpful to make the best of these data.

Quite right, we have now added a summarizing scheme (Figure 10).

The effect of treatments was quantified with 2D–outline analysis and statistics. Because they seem to produce opposite effects, samples with opposite treatments were grouped together. It would however be appropriate to also treat samples separately according to the treatment, even though significativity might not be reached because of limited sample size. It would be nice to point on the PCA the teeth that are shown later on pictures. Showing a superposition of outlines could also help the reader to get a general appreciation of treatments' effects.

We have now enhanced the PCA plot and Figure (Figure 5 – NOW FIGURE 6) to account for this, along with Reviewer 3’s suggestion.

This bias may result from accessibility of the drug through the tissue: This sounds weird to me. Why should it be a property of the treatment and not simply that the treatment reveals a property of the tissue?

Good point, we have modified the text to clarify this.

"Similarly, we observe the restricted expression of fgf3 (Figure 2F) and fgf10 (Figure 3A) very early during dental morphogenesis, prior to the establishment of the overall tooth shape". OK for Fgf3, but for FGF10, I could not find a picture where primary cusp morphogenesis is already ongoing??

We have now included a new Figure (Figure 2) to show the progression of expression in *fgf3*, fgf10 and shh from the earliest restriction of expression in the primary EK-like structure, through to later morphogenesis of the shark cusps. See Comment 2 and 6.

"EK signaling" seems to be used for "EK patterning" or "EK formation" at several places. Ex: "Wnt signalling appears to be a primary upstream regulator of EK signaling"

Thank you, we have checked and corrected this.

Reviewer #3 (Recommendations for the authors):I sympathize with the authors focus on Wnt signaling in the study, but somehow the argument why this focus was chosen is a bit mumbled. This raises the question why only Wnt signaling has been modified, and not, for example Shh which is known to produce comparable effects in the mouse (in fact, the unifying feature for the genes required for the progression of tooth development may be their high expression level, DOI: 10.1002/jez.b.23009). Moreover, it is pretty much to be expected that all epithelial organs are sensitive to modification of Wnt signaling. What I would suggest to slightly expand the introduction (Page 2, third paragraph starting "Whilst over a dozen markers have been described…"). A central point of canonical Wnt signaling in teeth is that it has different roles in epithelium and mesenchyme. A good reference outlining in its introduction why Wnts can be considered upstream in tooth development is https://doi.org/10.1242/dev.158048.

Thank you for the suggestion to add to our justification of the role of Wnt signalling in tooth development. We have added the following text to the relevant section of the introduction:

“The Wnt/B-Catenin signalling pathway appears to govern a variety of events during tooth development from initiation to morphogenesis. Canonical Wnt signalling is active in both major cellular contributions of the developing tooth, with varying roles in the mesenchyme and epithelium. For example, epithelial activation of Wnt/B-catenin signalling results in continuous initiation of new tooth units in mice, however, activation of Wnt signalling in the dental mesenchyme has an opposite effect, inhibiting sequential tooth emergence (Järvinen et al., 2018). Whilst over a dozen markers have been described within the mammalian EK, Wnt signalling appears to be a primary upstream regulator of EK patterning (Ahtiainen et al., 2016; Järvinen et al., 2018, 2006). Given this association with Wnt signalling and the control of tooth shape, including EK patterning, it seems appropriate to investigate further upstream effects of Wnt perturbation in alternative vertebrate models (Fraser et al., 2013; Richman and Handrigan, 2011).”

Additionally, we thank the reviewers for the hint related to the progression genes necessary for tooth development. We have added a small paragraph to the introduction to relate the key/core genes to the evolution of tooth shapes and conservation of the dental unit.

“Importantly, a unifying characteristic for the progression of vertebrate tooth development is the high level of cellular expression of developmental keystone genes (Hallikas et al., 2021). This suggests an essential core gene set (Fraser et al., 2010; Rasch et al., 2016) and their interactive signalling might be robust evolutionarily and highly conserved, capable of shape modifications of a conserved core vertebrate unit – the tooth”.

There are places containing inaccuracies about gene expression in mice. Most notably Shh is not only expressed in the enamel knots as currently stated (Page 6, "…shh is confined to the EK in mammalian molars…"). As the development proceeds, Shh expression spreads throughout the inner enamel epithelium (hence the wide use of ShhGFP mice to study tooth development). The results of Thiery et al. on Shh in shark teeth actually fit quite well with the dynamic spreading of Shh in the mouse. For example, see https://pubmed.ncbi.nlm.nih.gov/12403705/ or doi:10.1038/nature06153

We have added these citations and expanded the shh expression statement to include the spread of expression during morphogenesis: “In contrast, shh can be seen expressed within the dental epithelium and specifically upregulated within the leading edge of the tooth (Figure 5E and F). shh is confined to the EK in mammalian molars (Hardcastle et al., 1998; Vaahtokari et al., 1996), before spreading throughout the inner enamel epithelium later in tooth morphogenesis (Gritli-Linde et al. 2002; Kavanagh et al., 2007) and interestingly we see strong complementary expression within both the primary cusp (Figure 5E and F: black arrowhead) and later morphogenesis in the shark cusps (Figure 5E and F: white arrowhead; Figure 2). Depending on the stage of morphogenesis, the extent of shh expression within the inter-cusp dental epithelium is variable (See Figure 2).”

The discussion about Bmp4 expression is also a bit confusing. In the mouse, Bmp4 is only expressed in the enamel knot when those cells are about to enter apoptosis. It is true it is expressed in the mesenchyme adjacent to the EK and is required for the induction of the EK. As the shark teeth are much pointier than the mouse molars, it could still be that Bmp4 is involved in the induction of later forming cusps. Currently the interpretation is the opposite, and I leave it to the authors to decide how to address this (Page 4, "The precise exclusion of bmp4 from the non–proliferative tooth apex (Figure 3D and E) raises the possibility that instead of regulating putative EK signalling, it is involved in the restriction of other EK markers.").

We have added a sentence to this section to account for this interpretation: “However, we cannot rule out the involvement of Bmp4 in the induction of the later forming cusps.”

Not sure it is correct to call Msx a pathway. It is a transcription factor (homebox gene family) that integrates multiple pathways (Page 6 " Wnt/β–catenin signalling has been implicated upstream of key pathways (Bmp, FGF and Msx) regulating dental morphogenesis [12,13]")

Thank you – we have now changed this sentence to: " Wnt/β-catenin signalling has been implicated upstream of key signalling pathways (Bmp, FGF) as well as in the regulation of transcription factors (e.g., Msx) during dental morphogenesis [12,13]"

Wnt signaling manipulations: There are three places where the phenotypic results are stated to be 'dramatic' (pages 7, 12). Typically, when one needs geometric morphometrics to illustrate the results, the patterns are not very dramatic. In fact, looking at the PCA plot, the results are relatively subtle. I am also not following what is the difference between the main and supplement figure PCA (the main figure seems to be using a mean of the teeth)? Perhaps density ellipses for different treatments would help to illustrate the effects better, as they are there. And a more communicative adjective that could be used to describe the results is, for example, 'substantial'.

We have now replaced the two PCA figures with a single figure (Figure 6) that shows the cleared and stained tooth shapes to offer a visual of the shift in the morphology depicted by the PCA. Also, we have now changed/reduced the descriptor “dramatic” to the recommended adjective ‘substantial’.

The reference for the computational model using the ToothMaker is http://dx.doi.org/10.1098/rsos.180903This paper contains the used model version and the link to the github to obtain the package, and also provides the parameters for the basic tricusped seal teeth (Page 9, "After modelling a wild–type tricuspid tooth in correspondence with the software developers (J. Jernvall, pers. comm.)…")

Thank you – we have now added this reference in place of the pers. comm reference.

Finally, I wonder if a schematic illustration comparing mammalian and shark EKs would help to communicate the results, and also to point to issues that remain to be explored?

We agree and have now added a schematic diagram to the manuscript summarizing our comparison between shark and mammalian tooth and cusp development (Figure 10).

[Editors' note: further revisions were suggested prior to acceptance, as described below.]

The manuscript has been improved but there are some remaining issues that need to be addressed, as outlined below:1. In their reply to point 1, the authors focused strictly their reply on the suggestion to add experiments on the Shh pathway, but according to Reviewer 2 they did not reply to the first part. A few but mandatory text changes are necessary. This is detailed in the Reviewer 2 recommendations.

Please refer to the specific responses to Reviewer 2 (below).

2. There is a lack of support of the data showing that dental signaling centers are functionally similar, and the shark EK regulates the formation of cusps. The writing should be revised to tone down this and leave such conclusion for further functional studies. This should be revised and reworded according to reviewer 2 since the authors stated in their response and added this statement in the discussion (page 14, line 672) that "in the catshark at least, the site restriction of potential activatory and inhibitory signals of the oral epithelium may not affect the number of cusps….." Thus, the functional similarity in regulating the formation of cusps is not proven in the catshark and is not supported by the data.

Based on the comments related to this point by Reviewer 2 we have made appropriate changes to tone down the statements that suggest functional similarity.

3. The title should be changed as suggested by reviewer 2: "An epithelial signaling centre in sharks supports homology of tooth morphogenesis in vertebrates."

We thank Reviewer 2 for this suggestion; we have now changed the title as requested to “An epithelial signaling centre in sharks supports homology of tooth morphogenesis in vertebrates”.

4. The schematics 10 added by the authors should be revised according to the suggestion of reviewers 1 and 2, showing that the signalling centres in G and H reflect different structures in time and in position (for details see Prochazka et al., PNAS, 2010).

We have now added the reference to Prochazka et al., 2010 and a statement in the caption to account for the different structures in G and H. We have also coloured the mouse EK (G) pink to account for the difference compared to the shark, and the similarity between the mammalian EKs.

“G-I, Documented expression of genes in the pEK (G and H; although these two signalling centres in G and H represent different structures in time and place (as described by Prochazka et al., 2010))”

Reviewer #1 (Recommendations for the authors):Thank you for the revised version of the paper "An ancient dental signalling centre in sharks supports homology of tooth shape among vertebrates" submitted by Thiery et al.The authors revised their manuscript substantially according to the suggested points and in my point of view the revisions increased the quality of the manuscript substantially.I appreciate the change of the title of the manuscript, which now reflects much more the main focus of the paper and avoids confusion in homology of enamel knot in the mammals and enamel knot like structure in sharks.I also appreciate adding of the schematics as Figure 10, even if we would like to be precise, the situation of the mouse molar germ would probably from a sagittal view show a positional discrepancy between the signalling centres in G and H, since those two centres are not located in the same position of the epithelium, they differ in time and position as shown already by Prochazka et al., PNAS, 2010.Basically, all the addressed points have been sufficiently responded by authors and I do not have any additional comments.

Thank you for the comments and suggestions – the figure is shown with the shark schemes in both planes of section. Based on the comment from Reviewer 2 (below) we have now changed the figure to reflect the variation in the mouse EK, and have coloured the EK pink.

Reviewer #2 (Recommendations for the authors):Overall, the authors nicely take into consideration a number of points and I feel the manuscript has been improved. I have a major concern about a point that was not taken into consideration (1).1) About the general conclusion of the paperEssential revision point 1 was:"According to Reviewer 2 the data support the tip–down development of shark teeth with some critical involvement of the Wnt pathway more than the development of shark teeth from an apical signaling center. Thus an additional proof would be helpful to show if the gene expressions observed reflect strictly an existence of a signalling center as enamel knot in the mammalian word meaning, it means if both structures are really homological. The reviewers agree that it is expected that all epithelial organs are generally sensitive to modification of Wnt signaling. It is clear in this case, that obtaining new data is limited due to alternative model used in the study. However, please, consider a possible additional experimental focus on Shh pathway, what could be easily done since several inhibitors and activators are available. "In their reply to point 1, the authors focused strictly their reply on the suggestion to add experiments on the Shh pathway, but they did not reply to the first part. Therefore, I am not satisfied at all by the reply – this is not such a major issue however, since I believe only a few but mandatory text changes are needed to satisfy my concerns. This is detailed below.As already said in my review, the results obtained by the authors are very valuable, and surely an important step towards a recognition of some homology of tooth shape formation in vertebrates, but they surely do not definitely demonstrate that a signaling center works in sharks as it does in mammals.This inaccurate claim is based on an unappropriated use of modeling, which needs correction (see below). The experimental data clearly support a role of Wnt signaling in cusp development. The gene expression data are clearly in favor of tip–down development with an apical epithelial signaling center. The mammalian model with minor modifications can recapitulate shark different tooth morphologies and can (more or less) recapitulate morphological changes upon increased/decreased activation of the Wnt pathway. Some aspects are missed however (the recapitulation of Wnt–increased cusp number is far from being perfect). This tels us that shark teeth and their variation can be modeled with the developmental principles of mammalian teeth, and this further suggests that basic developmental principles of tooth morphogenesis are shared with sharks. But this cannot be taken as a definitive proof that an apical signaling center drives cusp formation as in mammals, nor that Wnt signaling takes part to this EK formation. This can surely be discussed in the discussion, but this is surely not a result from the model to be taken for granted. Overall, the data argue for common developmental principles between shark and mammals, including tip–down development with an apical signaling center, and suggest that Wnt signaling, as in mammals, may take part into activation–inhibition mechanisms responsible for the patterning of this signaling center. A plausible interpretation is not definitive proof, and it is important to keep the distinction for future research.

We thank the Reviewer (2) here for these insightful comments and fully agree with the interpretation. In addition to deleting or modifying text as suggested below, we have now added a caveat statement at the beginning of the discussion (as suggested):

“Overall, our data imply that a common set of developmental principles account for tooth morphogenesis between mammals and sharks. These common principles include the tip-down development of teeth from an apical signalling centre, and we suggest that Wnt signalling may take part in the activation-inhibition mechanisms that influence cusp patterning related to this signalling centre. Therefore, we suggest that the basic developmental principles of mammalian tooth morphogenesis are shared with sharks, and that perhaps shark tooth shape variation can be modelled with the same developmental principles. Future work will address whether there are indeed shared functional relationships between the mammalian and shark dental signalling centres.”

I recommend that several parts of the text are deleted (marked **…** below) or revised to stick the conclusions to the data. The text mentioned below should be removed, and the authors could expand the discussion of their results with more caution.As I will further detail below, one big issue in the reasoning is that there is no direct correspondence between model parameter and real–life signaling pathways – even in the mammal model. The tooth model is accounting for developmental principles, that’s all, and that’s already a lot.– P10. Line 480 implicate canonical Wnt in the regulation of cusp development **and therefore possibly in EK signalling**.Leave **…** for discussion.

Agreed. We have removed this statement.

– P11 Line 510–513 **Both properties could be reflective of 510 changes to underlying Wnt signalling, with increased activator/decreased inhibitor directly, or 511 a faster turnover of components favouring activation. Protein degradation is a key component 512 of Wnt signalling activity, of which the GSK3β complex which degrades β–catenin is just one 513 example**.These comments should surely not be not part of the result section. I strongly recommend to totally remove them: the tooth model has not the level of molecular realism that would make this comment meaningful. In mammals, many different pathways but the Wnt pathway have been linked to the activation/inhibition parameters of the model (Shh, Bmp4, Eda…), but no study pretends that one of this pathway "is the parameter". The above attempt to stick to molecular reality is artificial, and moreover, not accurate. Indeed, protein degradation is not taken by the authors as meant in the original model, which is inspired from Turing models = degradation should be the degradation of the activator when diffusing in space, not the degradation of an intracellular component of a signaling pathway.

Agreed. We have removed this statement.

P11 line 521–524 Thus, the shift from 5 to 6–cusps is more complex 521 than increasing the overall level of **Wnt** activation, simply doing so results in additional 522 enamel knots forming outside of the anterior–posterior axis. **These intricacies over shape 523 control are perhaps enabled by the complexity of Wnt signalling, the number of interacting 524 molecules permitting a wider range of phenotypes.**Again, these parts should surely not be not part of results, and should be removed in my opinion. There is a misunderstanding on what a model does and does not, and what a model can say or cannot say. The discrepancy between shark perturbed teeth and simulations should – first of all– be taken as an indication that the tooth model does not account for a key principle of shark tooth development – rather than it is missing the molecular complexity of a single pathway. Surely it does miss molecular complexity (that's what we ask to a model) – the question is, does it still capture some general principle (e.g. where additional cusp can form), or not. The authors should acknowledge and discuss more appropriately the discrepancy in the discussion part.

Agreed. We have therefore deleted the suggested text:

“Thus, the shift from 5 to 6-cusps is more complex than increasing the overall level of activation, simply doing so results in additional enamel knots forming outside of the anterior-posterior axis.

Additionally, we have added a statement in the first paragraph of the discussion to account for the model versus real biology:

“Of course, care should be taken when interpreting the results of a model compared to real biological signalling, where a model cannot simply account for the complexity of a single pathway.”

**In the model, the 550 regulation of the inhibitor and activator stems from the EK ignaling centre (Salazar–Ciudad and Jernvall, 2010). The recreation of comparable phenotypes through the alteration of biologically relevant genetic MODEL parameters supports our findings, which demonstrate a critical role of canonical Wnt ignaling in the regulation of cusp number from the catshark EK (Figure 9). **No, I strongly disagree with this conclusion for the reasons mentioned above. This text should be modified.

We have modified this statement to tone down the conclusion, and suggest care when implying model data:

“In the model, the regulation of the inhibitor and activator stems from the EK signaling centre (Salazar-Ciudad and Jernvall, 2010). The recreation of comparable phenotypes through the alteration of biologically relevant model parameters, suggests a potential role of canonical Wnt signaling in the regulation of cusp number in the catshark (Figure 9).”

Line 560–568Overall, our results demonstrate that signalling centres comparable to mammalian enamel knots are present in the shark during tooth cusp morphogenesis. **Our data implies that these dental signalling centres are functionally similar in both sharks and the mammals. The enamel knot (EK)–like signalling centre regulates the formation of dental cusps in the catshark with canonical Wnt signalling playing a significant role in this process. **Again, I disagree that the data show that dental signaling centers are functionally similar, and the shark EK regulates the formation of cusps. The writing should be revised to tone down this and leave such conclusion for further functional studies.

We agree and have removed this statement on the comparable functionality.

“Overall, our results demonstrate that signalling centres comparable to mammalian enamel knots are present in the shark during tooth cusp morphogenesis. Our data implies that these dental signalling centres function to generate cusps in the shark. Although, how functionally similar these are to mammalian EKs requires further investigation.

2) About the new title:"An ancient dental signalling centre in sharks supports homology of tooth shape among vertebrates"Is it really a matter of tooth shape homology? – I guess the focus is rather the homology of developmental principles? It is unclear to me what for "ancient" stands for: "old like early vertebrate teeth" or "older than teeth themselves". In the first case, it is redundant with the idea of homology in vertebrates, in the second case, I am not sure that the two ideas can be conveyed at the same time in the title. For all those reasons, I would suggest something simple like:– An epithelial signaling centre in sharks supports homology of tooth morphogenesis in vertebrates

As above, we agree and have changed the title to the suggested statement: “An epithelial signaling centre in sharks supports homology of tooth morphogenesis in vertebrates”.

3) About the revised section on gene expressionPlease consider improving the writing of the "Key markers of tooth morphogenesis in sharks" section. This long description lacks clear organization for me – I missed the line. For example : Fgf3, Fgf10, Shh are introduced lines 191–198, but fgf3 and fgf10 are discussed again l 237–251, between a Wnt– and a Bmp4– paragraph. At the very end of the section, these 3 genes are mentioned again with this statement, which sounds like a repetition:" fgf3, fgf10, sonic hedgehog (shh)midkine (mdk) have previously been observed within the apical tip of developing teeth in the catshark (Scyliorhinus canicula) and little skate(Leucoraja erinacea), corresponding to the putative EK–like signalling centre (Rasch et al.,299 2020, 2016).There might be a good reason why mdk is discussed in this very last part, but at least revise the text to put the focus on mdk: “mdk has been previously associated with fgf3, fgf10 and shh in”…"Same thing, line 30”: "Another marker, isl1, is expressed (Figure 3P) in a very similar and restricted pattern to fg”10".It would really help me if all these overlapping gene expression patterns are mentioned together in the same paragraph.I found figure numbering quite confusing and it seems to me it does not correspond to the text order. Why are Bmp4 results (coming very late in the section) associated to Figure 2 (fgf3,fgf10,shh)?

Thank you for these useful comments. We agree with the comment on mdk and isl1 and have changed the text to justify the description of these markers, and removed any repetition of FGF and shh descriptions. We have also moved these descriptions of gene expression to a more relevant portion of the section to relate better to the FGF text.

We have also moved the bmp4 description section, based on the reviewer's comment, to a more appropriate section that now makes sense in association to Figure 2 as a supplement documenting a similar stage of the first generation teeth. Thank you for this suggestion to clarify the text.

4) About the new carton/figure 10:The younger cartoon represents a mouse molar bud stage, just before cap transition. At this stage, the first molar EK has just formed and is about to drive cap transition. Bmp4 is expressed in the PEK at this stage (Prochazka et al. 2010).Therefore, given the color code, the mouse EK should thus be pink, and not red.

We agree, thank you. We have now changed the colour of the mouse EK to pink.

Reviewer #3 (Recommendations for the authors):I find the revised manuscript much improved. While one could pick on individual parts, the cumulative evidence provided by the work is compelling. Moreover, the work should stimulate many new studies, both experimental and comparative. I like the 'apical epithelial knot' term proposed.Below a few details to be fixed.– For consistency, Sostdc1 should be used instead of Ectodin for the gene name (both mentioned in the text but ectodin used in figures). In fact, Sostdc1 has also names 'wise' and 'Usag1'.

Agreed, and changed.

– Supplementary Table S1 should refer to Savriama et al. 2018. Incidentally, I noticed that quite a few parameters have been tinkered with to produce the 'seal' morphology. Although some changes were quite likely unnecessary, I checked and the result is a seal morphology.

Reference added to supplemental table S1.

– Reference list should be checked. At least Hallikas et al. 2021 is missing.

We have checked the reference list, and made relevant adjustments.

– Figure 6 (tooth shape analyses) is challenging to read, especially with the very long caption. There are color codes for the different treatments, but especially Figure 6D (PCA plot) is confusing as one tooth is circled with each color. How about trying to use colored 95 or 99% density ellipses for each treatment? Moreover, the Figure 6A has red–green color code that is confusing as these colors do not mean the same as in the rest. Is the 6A needed?

We have now changed Figure 6 based on these suggested edits and we feel it is easier to follow (ellipses added, and the original 6A removed) and the separated/removed section has been added to a supplemental part. Additionally, the figure citations have been changed to reflect this modification.

In addition: We have also added code availability through Github for reproducibility (information added to the manuscript file).

Code Availability

Our analysis code, plots and morphometrics data can be found and re-ran at https://github.com/alexthiery/scanicula-aek. We have also included a Docker container for reproducibility docker://alexthiery/scanicula-aek:v1.0.